# Dolomite luminescence thermochronometry reconstructs the low-temperature exhumation history of carbonate rocks in the central Apennines, Italy

Junjie Zhang [1] ✉, Giorgio Arriga [2,3], Federico Rossetti [2] ✉, Valentina Argante[1], Dennis Kraemer [4], Mariana Sontag-González [5], Domenico Cosentino [2], Paola Cipollari [2] & Sumiko Tsukamoto[1,6]

The lack of available thermochronological methods has so far hampered reconstructions of the cooling and exhumation histories in carbonate rock regions. Here we develop a new trapped charge thermochronometry tool based on the thermoluminescence signal of dolomite. It has a closure temperature range of 45–75 °C and is applicable to carbonate domains with cooling rates of 2–200 °C per million years. This new thermochronometric technique is tested in the central Apennines, where seismogenic, carbonate-hosted normal faulting controls regional neotectonics. Thermoluminescence dating is applied along the northeastern shoulder of the Late Pliocene-Quaternary L'Aquila Intermontane Basin, at the footwall of the extensional Monte Marine Fault. Dolomite samples from the bedrock have a mean thermoluminescence age of 4.60 ± 0.35 millions of years, whereas dolomite clasts within the fault damage zone have a mean thermoluminescence age of 2.53 ± 0.13 millions of years. These new thermoluminescence ages, corroborated by the existing stratigraphic constraints, (i) provide the first direct, low-temperature exhumation ages of the carbonate bedrocks in the central Apennines; (ii) constrain the activity of the basin boundary faults along the northeastern shoulder of the L'Aquila Intermontane Basin. Our study demonstrates the potential of dolomite luminescence thermochronometry in reconstructing the low-temperature cooling/exhumation history of carbonate bedrocks.

Reconstructing the cooling and exhumation histories of carbonate bedrocks has long been challenging due to the inapplicability of the classical low-temperature thermochronometry techniques, such as zircon and apatite fission-track and (U–Th)/He methods[1]. (U-Th)/He dating has been explored for carbonates; however, its application is hindered by the low concentrations of U and Th, as well as the complex diffusion kinetics of helium within carbonate minerals[2,3]. This is a major limitation for the tectonic reconstruction in regions where carbonate bedrocks constitute the backbone of the exhumed terranes, such as in the Alpine circum-Mediterranean orogens[4].

In recent years, luminescence thermochronometry has been developed and applied to unravel the cooling histories of rock bodies[5–8]. Luminescence thermochronometry is based on the competition between the trapping of free charges (induced by ionizing irradiation) into the electron traps inside the mineral lattice (defects and impurities) and the thermally stimulated detrapping (with the contribution of fading for feldspar) of the charges from the traps. The concentration of trapped charges in a certain mineral can be measured either as optically stimulated luminescence (OSL) or as thermoluminescence (TL), which are evicted by light and heat, respectively.

[1]Leibniz Institute for Applied Geophysics (LIAG), Hannover, Germany. [2]Dipartimento di Scienze, Università Roma Tre, Rome, Italy. [3]Archaeology, Environmental Changes, and Geo-Chemistry, Vrije Universiteit Brussel (VUB), Brussels, Belgium. [4]Federal Institute for Geosciences and Natural Resources (BGR), Hannover, Germany. [5]Institute of Geography, Justus Liebig University of Giessen, Giessen, Germany. [6]Department of Geosciences, University of Tübingen, Tübingen, Germany. ✉e-mail: Junjie.Zhang@leibniz-liag.de; federico.rossetti@uniroma3.it

The growth of the luminescence signal with irradiation follows a single saturating exponential function[9], as

$$I_n = I_0 * (1 - e^{-(D/D_0)}) \qquad (1)$$

where $I_n$ is the natural luminescence intensity, $I_0$ is the maximum luminescence intensity at saturation, $D_0$ is the characteristic saturation dose, and $D$ is the dose the mineral received in nature. In dating, $D$ is expressed as equivalent dose ($D_e$). The luminescence age can be calculated by the ratio $D_e/\dot{D}$, where $\dot{D}$ is the environmental dose rate (deduced mainly from U, Th, K concentrations). The effective closure temperature ($T_C$) of luminescence thermochronometry ranges from ~30 °C to ~90 °C, depending on the mineral and signal, as well as the cooling rate and dose rate[10,11]. The most used minerals for luminescence thermochronometry are quartz and feldspar. Since the number of electron traps in the mineral lattice is not infinite, the traps will be fully occupied with a high amount of irradiation, i.e., the luminescence signal gets saturated. The low dating limits (low $D_0$ values) of quartz and feldspar make their luminescence thermochronometry only applicable to rapidly exhuming terrains, with cooling rates higher than 200 °C Ma$^{-1}$ [8]. The concentration of trapped charges in quartz can also be measured with the electron spin resonance (ESR), which has a higher dating limit of ~2 Ma, so that the quartz ESR signal can recover cooling rates as low as 20–50 °C Ma$^{-1}$ [12–14]. In regions with cooling rates lower than 200 °C Ma$^{-1}$ (with regard to luminescence) or lower than 20 °C Ma$^{-1}$ (with regard to ESR), the luminescence signals of quartz and feldspar and the ESR signal of

quartz in rocks exhumed to the surface are already in saturation. Furthermore, quartz and feldspar trapped charge thermochronometry is limited to rocks containing these minerals, and thus not applicable to carbonate rocks.

In this study, we develop the dolomite TL thermochronometry as a new tool to assess the cooling and exhumation histories of carbonate rocks. Diagenetic and/or structurally-controlled dolomitization is widely documented in carbonate bedrocks, imparting a first-order control on the mechanical response and permeability distribution in carbonate rocks[15–20]. Compared to dolomite, calcite is the more abundant carbonate mineral. The semi-stable 230 °C TL peak of calcite is sensitive to surface temperature and the thermal equilibrium state of this TL signal has been thus applied to reconstruct the past surface temperature on Earth[21,22]. In contrast, dolomite has a strong TL peak above 300 °C[23–25], which has much higher thermal stability at the surface temperature. Previous studies on the luminescence characteristics of dolomite show that its TL signal has a strikingly high $D_0$, up to thousands of grays[23–25]. Such a high $D_0$ ensures a high dating limit up to tens of million years, which would be a significant advantage in reconstructing thermal histories in regions with low cooling rates. Since dolomite rheology has been documented to be harder than calcite during brittle deformation[26,27], dolomite TL thermochronometry may potentially provide a useful tectonic/thermal marker to investigate the long-term crustal deformation in carbonate bedrocks.

We test the potential of the dolomite TL thermochronometry in reconstructing the cooling/exhumation history of the carbonate seismogenic crust of the central Apennines (Italy), in the L'Aquila region (Fig. 1a).

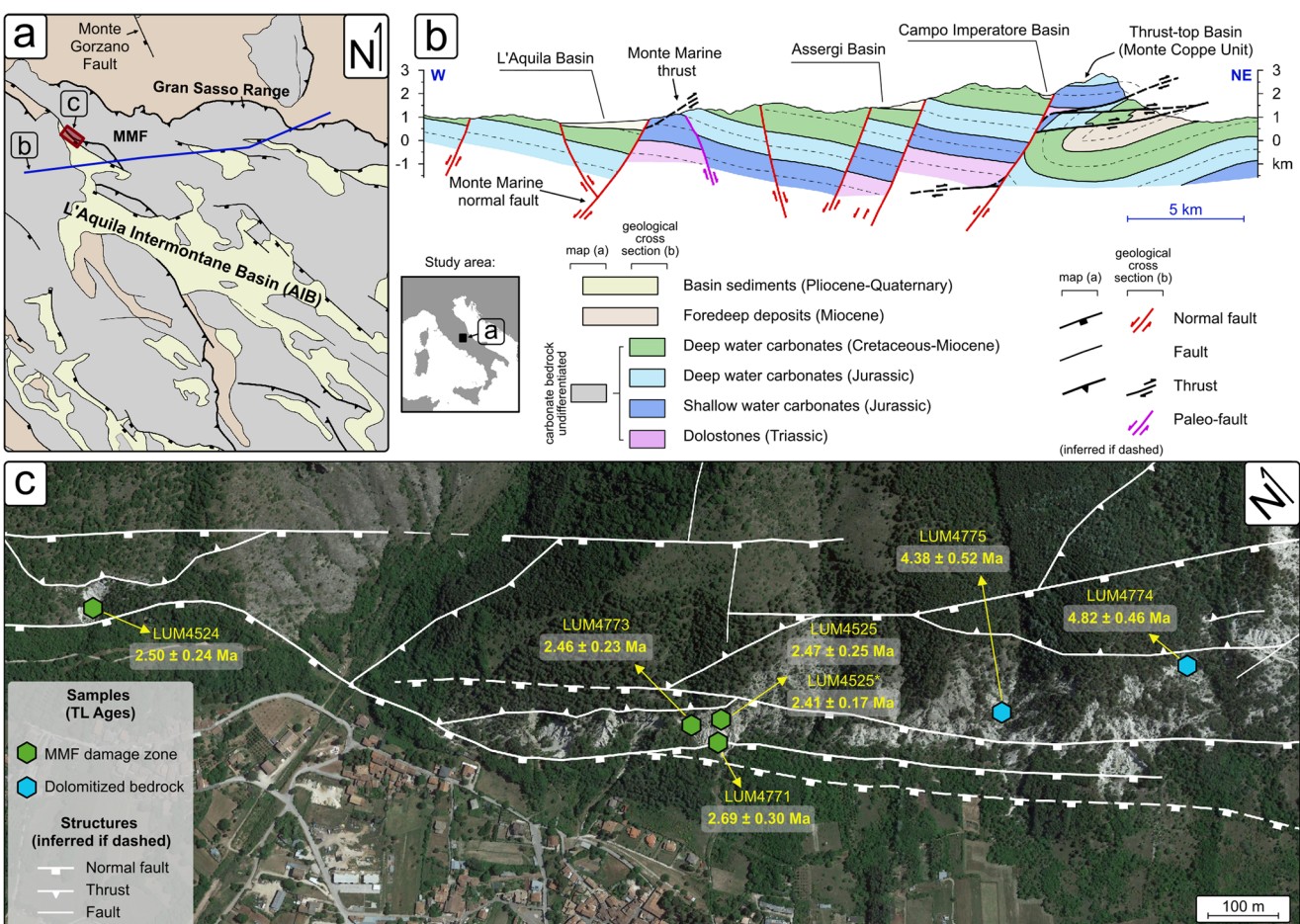

**Fig. 1 | The study area framed within the regional geology of the central Apennines. a** Simplified geological scheme of the central Apennines, showing the main tectonic structures and the main paleotectonic domains. MMF: Monte Marine Fault. **b** Geological cross section across the central Apennines (from the L'Aquila Basin to the Gran Sasso Range) showing the general structural architecture of the range (based on the CARG sheet n° 349-Gran Sasso d'Italia[112] and n° 348-Antrodoco[113]. **c** Satellite image of the study area (Map data: Google earth, Airbus), showing the main fault traces and locations of the studied samples, including the corresponding dolomite TL ages (see also Table 1).

The central Apennines provide a natural laboratory to study the long-term response of the brittle upper crust to regional tectonics through dolomite thermochronometry, since pre-orogenic dolomitized carbonate rocks[28–32] are involved in Neogene-Quaternary syn-orogenic crustal shortening and post-orogenic crustal extension. The central Apennine orogen consists of a NE-verging thrust-and-fold belt formed by the Late Miocene to the Early Pliocene stacking of Mesozoic-Cenozoic carbonate successions (Triassic-Neogene platform-basin system[33]). Since the Late Pliocene, post-orogenic extension affected the backbone of the orogen, causing the formation of intermontane basins, such as the L'Aquila Intermontane Basin (AIB, Fig. 1a), bordered by NW-SE striking high-angle normal faults[34,35] (Fig. 1a, b). These extensional fault systems are responsible for the present seismotectonic regime of central Italy, characterized by earthquake sequences of moderate to high intensity, and potentially capable of earthquakes with $M_w > 6.5$[36–41].

Despite a good knowledge of the regional seismicity and the neotectonic setting of the central Apennines, still poorly known is the exhumation history of the carbonate bedrock during the transition from syn- to post-orogenic tectonics. The orogenic crustal shortening regime (stage $D_1$) has been attributed to the Early Pliocene (~5.5–5.0 Ma), as deduced from the stratigraphic age of the thrust-top basins at the leading edge of the orogenic chain (e.g., Le Vicenne and Monte Coppe thrust-top basins[34,42–44]). The onset of the post-orogenic extension (stage $D_2$) can be deduced by the stratigraphy of the syn-rift deposits. Based mainly on magnetostratigraphy and biochronology, the basal infill of the AIB has been referred to the Late Pliocene (Piacenzian, 3.60–2.58 Ma[45]), or the late Piacenzian–early Gelasian (ca. 3.0–2.0 Ma[46]), or the Early Pleistocene (Gelasian, 2.58–1.8 Ma[47]). Direct age of the $D_2$ extensional faulting in Central Apennines has been obtained through U-Pb dating of calcite mineralizations associated with the Monte Gorzano Fault (Fig. 1a), showing that extensional deformation began at least ~2.5 Ma ago[48]. Finally, geomorphological, geochemical and fission track thermochronological results from the central Apennines have documented a major erosion/exhumation event since ca. 3.0–2.5 Ma during the early stages of formation of the intermontane basins, at rates of ~1 mm yr$^{-1}$ [49–52].

The Monte Marine Fault (MMF) is part of the northeastern basin boundary faults of the AIB, a 50–60 km long extensional fault system, able to generate earthquakes with $M_w$ higher than 6.5[53–57] (Fig. 1a). The MMF comprises several active extensional fault strands with en-echelon geometries that, striking NW-SE and dipping to the SW, form a ca. 10 km-long deformation zone crossing the lower piedmont of the Monte Marine[37,58]. These fault segments cut across previously deformed, Lower Jurassic – Upper Cretaceous carbonate bedrock and the Upper Miocene foredeep siliciclastic deposits[32]. The Lower Jurassic carbonate succession (Calcare Massiccio Fm and Corniola Fm) documents Jurassic rifting, with the formation of the Monte Marine Pelagic Carbonate Platform[59–62]. The Monte Marine Pelagic Carbonate Platform shows evidence of post-sedimentary dolomitization[32], similarly to what was observed in analogous carbonate platform systems of the central Apennines[28–31]. The dolomitized carbonates are affected by two main stages of Neogene to Quaternary regional deformation[58]: (i) $D_1$, orogenic shortening; and (ii) $D_2$, post-orogenic extension (Fig. 2). The $D_1$ deformation developed in response to a NE-SW-directed maximum compression direction, with formation of disjunctive and anastomosed solution cleavage domains and low-angle cataclastic zones (meter thick) along major NE-verging thrust systems in the carbonate bedrocks (Fig. 2a, b). The $D_2$ deformation overprints the $D_1$ structures and reworks the thrust-related cataclastic zones along sub-vertical fault zones downfaulting the thrusted units to the SW along the MMF strands (Fig. 2a, c). The damage zone of the MMF is characterized by a large volume of in-situ shattered rocks hosting dolomite clasts of different sizes (Fig. 2d), which are interpreted as the product of multiple seismogenic events in a mechanically heterogeneous carbonate substratum[58]. Therefore, pre-orogenic dolomite grains (preserved either as carbonate bedrock or as fault clasts within the damage zone of the MMF) may provide an ideal strain marker to reconstruct the tectono-thermal history of the Apennine carbonate bedrock from $D_1$ orogenic contraction to $D_2$ post-orogenic extension.

## Results and discussion
### Dolomite TL ages and cooling rates
Four dolomite clasts were collected within the damage zone of the extensional fault at the piedmont of the Monte Marine, and two dolomitized bedrock samples were collected at different distances (~20 m and ~100 m, respectively) from the main fault trace (Fig. 1c; Table S1; Methods section). X-ray diffraction (XRD) analyses indicate that the samples are made of almost pure dolomite with only trace amount of calcite (Fig. S1). TL spectral measurements show that the emission is centered at ~580 nm for both the bedrock and clasts (Figs. 3a and S2). Natural TL curves have two peaks at ~280 and 350 °C with a heating rate of 5 °C s$^{-1}$, while TL curves with artificial dose administered in laboratory show three peaks at ~150 °C, 250 °C and 350 °C, respectively (Fig. 3b and S3). Peak deconvolution indicates another peak at ~295 °C (Fig. S3). The 280 °C TL peak in the natural signal appears to be a combination of the two TL peaks at ~250 and 295 °C. In thermochronometric application, the TL curves with laboratory doses were measured after a preheat to 260 °C to remove low-temperature peaks, and the 350 °C TL peak is of our interest. A multiple-aliquot additive-dose dating protocol was used[63] (Table S2). The TL intensities of different groups of aliquots with different additive doses (0–8000 Gy) were measured and the relationship between the TL intensity and the additive dose was fitted with Eq. (1) to obtain the $D_e$ (Figs. 3c, d, S4). The TL intensity with 0 Gy additive dose is the natural TL signal ($I_n$). Meanwhile, the maximum TL intensity at saturation ($I_0$) can be deduced from the fitting. The signal saturation level of $I_n/I_0$ will be used for the cooling history modeling. From the $D_e$ values and dose rates, the two dolomitized bedrock samples (LUM4774, LUM4775) have apparent TL ages of $4.82 \pm 0.46$ and $4.38 \pm 0.52$ Ma (1σ error), respectively, with a mean age of $4.60 \pm 0.35$ Ma. The four dolomite clasts (LUM4524, LUM4525, LUM4771, LUM4773) on the fault damage zone provide apparent TL ages of $2.50 \pm 0.24$, $2.47 \pm 0.25$, $2.69 \pm 0.30$, and $2.46 \pm 0.23$ Ma, respectively (Table 1). The mean TL age of the dolomite clasts is $2.53 \pm 0.13$ Ma, which is ~2.1 Myr younger than the dolomitized bedrocks.

For trapped charge thermochronometry, the cooling rate can be deduced by modeling the charge accumulation with the cooling history. For minerals without anomalous fading, the rate of charges accumulating inside the electron trap responsible for the luminescence signal can be described by the following equation[6,8,64], with the first-order kinetics:

$$\frac{d(\frac{n}{N})}{dt} = p_{trapping}\left(1 - \frac{n}{N}\right) - p_{detrapping}\frac{n}{N} = \frac{\dot{D}}{D_0}\left(1 - \frac{n}{N}\right) - \frac{1}{\tau}\left(\frac{n}{N}\right) \quad (2)$$

where $t$ (s) is time, $n$ is the number of occupied electron traps, $N$ is the total number of the electron traps, $n/N$ is the trap saturation level which is represented by the luminescence signal saturation level ($I_n/I_0$). The $p_{trapping}$ is the probability of a charge occupying the trap, which equals to $\dot{D}/D_0$. The $p_{detrapping}$ is the probability of a charge escaping the trap, which equals to $1/\tau$. $\tau$ is the thermal lifetime of the trapped charges, which can be described by the following equation:

$$\tau = s^{-1}e^{E/kT} \quad (3)$$

in which $E$ (eV) and $s$ (s$^{-1}$) are the activation energy and the escape frequency factor of the trap, respectively, $T$ (K) is the temperature, and $k$ is the Boltzmann constant.

With a certain cooling rate, the signal growth with time (with a resolution of 1 kyr) can be modeled from an initial temperature of 100 °C (so high temperature that electrons cannot be stored in the traps) to the surface temperature (10 °C in L'Aquila). Since the signal saturation level of a sample at the surface can be measured by $I_n/I_0$, its cooling rate can thus be deduced (Fig. 4a, b). The cooling rates of the two dolomitized bedrock samples are $11.3 \pm 2.2$ and $11.3 \pm 1.7$ °C Ma$^{-1}$ (1σ error), respectively, which are identical to each other. The effective closure temperature ($T_C$) is a key parameter for any thermochronometry method. Here, we calculate the $T_C$ values from the TL ages, the surface temperature ($T_S = 10$ °C) and the cooling rates of the

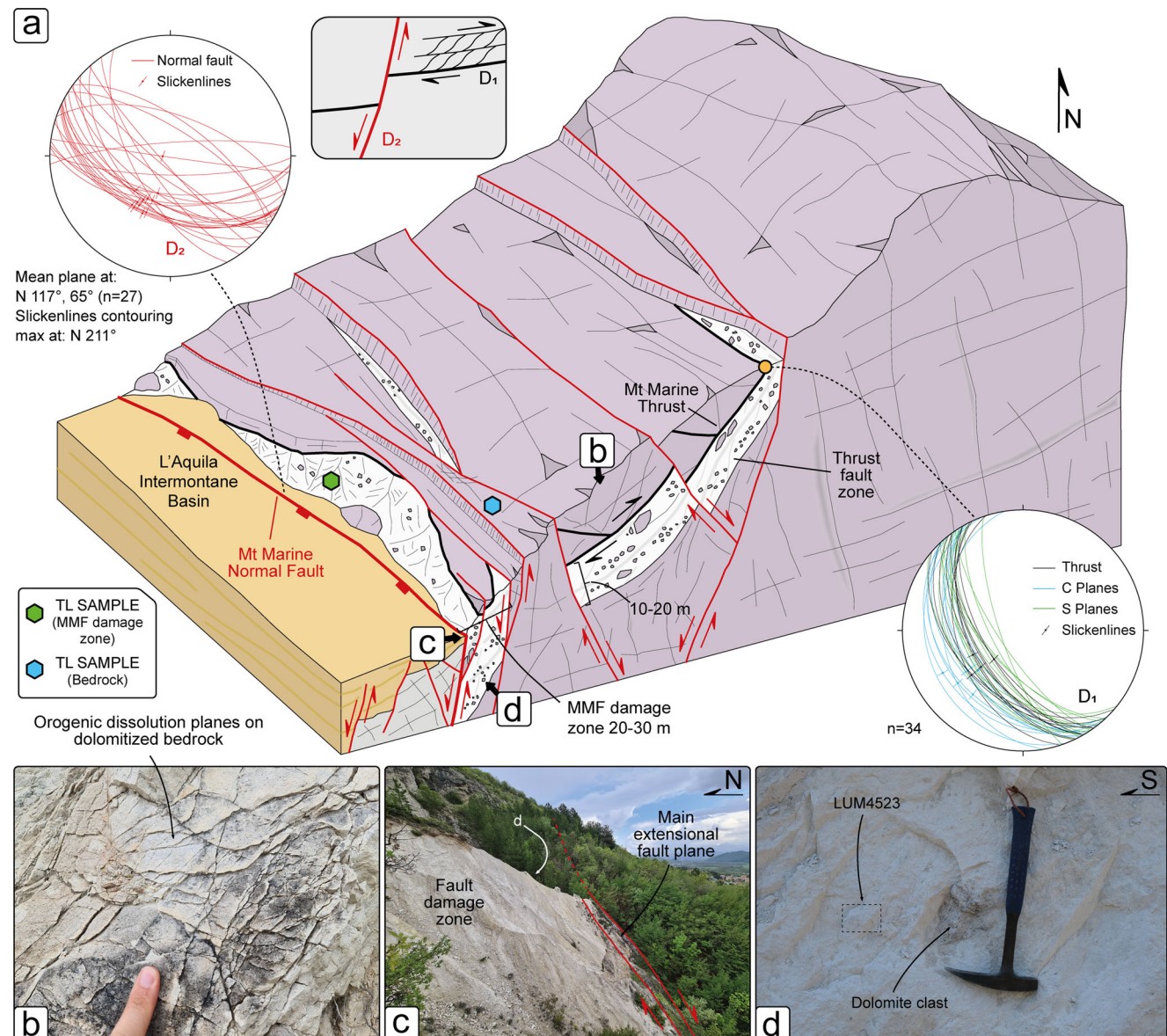

**Fig. 2 | The structural architecture at the piedmont of the Monte Marine. a** 3D structural scheme of the study area (not to scale, location of structures is only indicative) showing the tectonic overprinting of $D_2$ post-orogenic extensional faulting on the $D_1$ contractional structures. The stereoplots (Schmidt lower emisphere projection) shows representative $D_1$ and $D_2$ structures. **b** $D_1$ anastomosed dissolution surfaces formed by tectonic shortening in the dolomitized bedrock, indicating the pre-orogenic origin of dolomitization. **c** Distributed extensional faulting within the damage zone of the Monte Marine Fault (MMF) $D_2$ at the piedmont of the Monte Marine. **d** Detail showing dolomite clasts within the MMF rocks.

two bedrock samples (Table 1), using the equation:

$$Cooling\ rate = (T_{C_-}T_S)/age \qquad (4)$$

The $T_C$ values of the two bedrock samples are 64.5 and 59.5 °C, respectively. Within the cooling rate modeling, these $T_C$ values correspond to a detrapping/trapping rate ratio of 50% for LUM4774 and 51% for LUM4775. Thus, we can also assume that the $T_C$ is the temperature at which the electron detrapping rate is half of the trapping rate (i.e., 50% open system) during the cooling history (Fig. 4c).

With trapped charge thermochronometry, the $T_C$ changes with many parameters, i.e., cooling rates, $\dot{D}$, $D_0$ and thermal kinetic parameters ($E$, $s$) of the trap[10,11]. For the 350 °C TL peak of six dolomite samples in this study, the $E$ varies in the range of 1.71–1.88 eV, $\log s$ in the range of 13.0–14.6, and $D_0$ in the range of 2450–5920 Gy (Table 1). To study the variation of the effective closure temperature at different cooling rates and environmental dose rates,

we applied a median $E$ of 1.80 eV, a median $s$ of $1 \times 10^{14}\,s^{-1}$ and a median $D_0$ of 5000 Gy, to simulate the signal growth with the cooling rate in the range of 2–200 °C Ma$^{-1}$ and the $\dot{D}$ in the range of 0.1–2.0 Gy ka$^{-1}$. Under each pair of cooling rate and $\dot{D}$, the signal saturation level of a dolomite sample exhumed to the surface can be obtained, which can be converted into a TL age from Eq. (1). Consequently, a $T_C$ can be calculated from Eq. (4). Overall, the $T_C$ is in the range of ca. 45–75 °C (Fig. 4d). The $T_C$ is sensitive to the cooling rate, with a higher $T_C$ at a higher cooling rate. However, it is not sensitive to the dose rate except when the cooling rate is very low (<5 °C Ma$^{-1}$).

The younger TL ages of the dolomite clasts (~2.1 Myr younger than the bedrocks) in the fault damage zone of the MMF is the evidence that processes other than rock exhumation have operated to reset the dolomite TL signal, such as (i) a fluid-mediated (re-)crystallization and/or (ii) frictional heating during fault activity[65–70]. The shale-normalized rare earth elements and yttrium (REY) of the dolomite samples (bedrock, fault clasts and cataclastic matrix) overlap with each other, showing a consistent REY signature (Fig. S5), with anomalies that are typical for seawater-derived chemical

## a) TL emission spectra

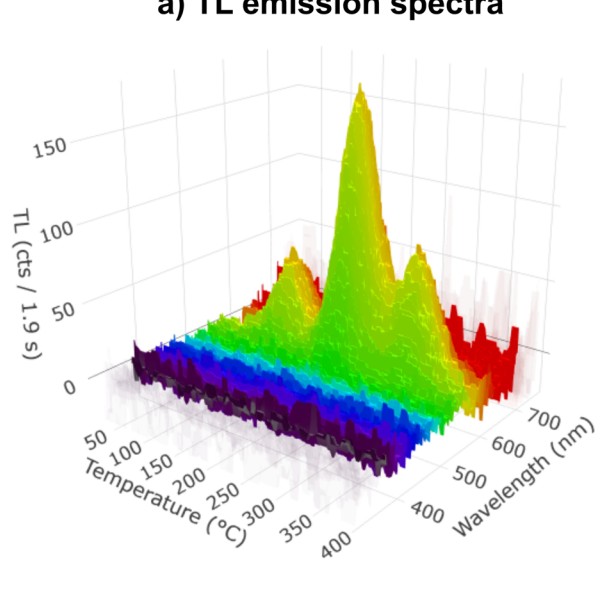

## b) TL curves

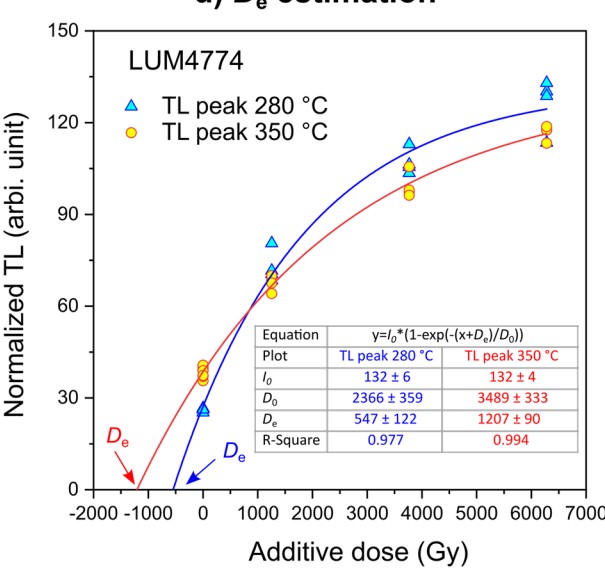

## c) TL curves for $D_e$ measurements

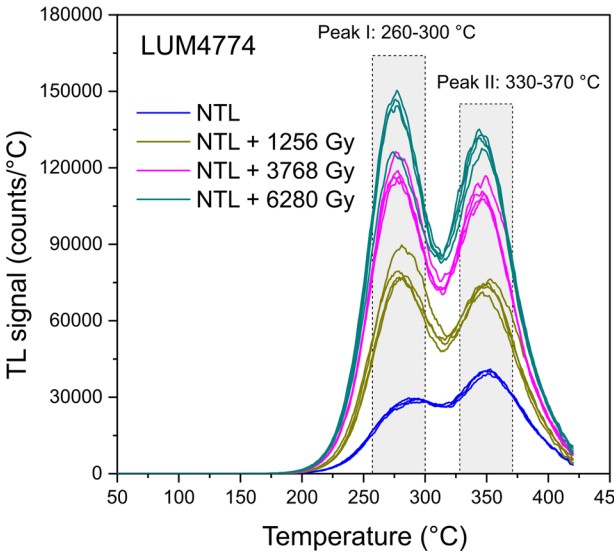

## d) $D_e$ estimation

**Fig. 3 | Thermoluminescence emission spectra and $D_e$ measurements. a** TL spectra of a dolomitized bedrock sample (LUM4774). The signal corresponds to a regenerative dose of ~1000 Gy. The emission is peaked at 580 nm for all three TL peaks. The heating rate is 1 °C s$^{-1}$. **b** The TL curves of a clast sample LUM4524. The heating rate is 5 °C s$^{-1}$. RTL is the TL signal corresponding to a regenerative dose of 2340 Gy. RTL (after PH) is the regenerative-dose TL signal after a preheat treatment to 260 °C. NTL is the natural TL signal. NTL curves of 5 aliquots are shown. NTL

(after PH) is the natural TL signal after a preheat treatment to 260 °C. **c** TL curves for the natural signals (NTL) and the natural + additive dose signals for LUM4774. The TL curves with added doses were recorded after a preheat treatment to 260 °C to remove the low-temperature TL signal. Four aliquots were used for each group. The heating rate is 5 °C s$^{-1}$. **d** Dose response curve fitting and $D_e$ estimation with the multiple-aliquot additive-dose protocol. The fitted function is modified from Eq. (1): $I_n = I_0 * (1 - e^{-(x+D_e)/D_0})$.

sediments, such as positive La, Gd and Y anomalies and a negative Ce anomaly[71,72]. Variations in the overall REY concentrations among the different samples are thus probably related to regional variations within the lithological units, as the sample sites are hundreds of meters apart. Indeed, previous studies have reported variations of REY concentrations up to three orders of magnitude for Italian limestones[73]. Therefore, the similarity of the REY fractionation patterns of the studied samples suggests a closed chemical system without significant alteration/metasomatism in the carbonate bedrock during faulting. Accordingly, the more feasible scenario for the TL signal resetting appears to be the frictional heating during the fault slip. The luminescence and ESR signal resetting due to frictional heating have been reported for fault zone samples[74–76] and validated by laboratory experiments[65,66,77–79]. The extent to which luminescence and ESR signals are reset, completely or partially, depends on factors such as the slip rate, stress

regime, and proximity to the fault core[65,66,79]. Since polyphase and active seismogenic faulting is documented along the MMF since Pleistocene[37,58,80,81], the apparent TL ages of ~2.5 Ma of the fault clasts might be a combined result of several partial resetting events that accumulated over the exhumation history of the clasts. However, under this scenario, the similar apparent TL ages of the fault clasts would require similar degrees of partial resetting between the clasts during these events. Considering the different sizes and different along-strike locations of the clasts which should result in different susceptibilities to frictional heating, the scenario of partial resetting with similar degrees between different clasts is unlikely to occur. Instead, the similar TL ages of the clasts can be reasonably explained by a complete resetting scenario, which can be achieved by a single seismic event or multiple seismic events within a seismic phase, as long as the frictional heating is sufficiently intense. With the same slip rate, the luminescence

**Article**

**Table 1 | Sample information, TL ages (for the 350 °C TL peak) and modeled cooling rates for dolomite samples**

| LUM ID | Sample ID | Sample type | Ḋ (Gy ka⁻¹) | $D_0$ (Gy) | $D_e$ (Gy) | Age (Ma) | n/N | E (eV) | logs (s⁻¹) | Cooling rate (°C Ma⁻¹) | $T_C$ (°C) |
|---|---|---|---|---|---|---|---|---|---|---|---|
| LUM4524 | MAR-2-TL | Fault clast | 0.849 ± 0.056 | 4625 ± 486 | 2123 ± 145 | 2.50 ± 0.24 | 0.371 ± 0.013 | 1.767 ± 0.017 | 13.45 ± 0.10 | | |
| LUM4525 | MAR-4-TL | | 0.669 ± 0.037 | 5918 ± 890 | 1650 ± 142 | 2.47 ± 0.25 | 0.241 ± 0.018 | 1.783 ± 0.004 | 13.63 ± 0.03 | | |
| LUM4525ᵃ | MAR-4-TL | | 0.669 ± 0.037 | 5561 ± 405 | 1612 ± 68 | 2.41 ± 0.17 | 0.250 ± 0.008 | 1.783 ± 0.004 | 13.63 ± 0.03 | | |
| LUM4771 | MAR-7-TL | | 0.491 ± 0.026 | 3734 ± 624 | 1320 ± 129 | 2.69 ± 0.30 | 0.294 ± 0.017 | 1.771 ± 0.030 | 13.62 ± 0.27 | | |
| LUM4773 | MAR-18 | | 1.024 ± 0.055 | 5494 ± 691 | 2520 ± 198 | 2.46 ± 0.23 | 0.369 ± 0.025 | 1.710 ± 0.022 | 12.99 ± 0.19 | | |
| LUM4774 | MAR-29 | Bedrock | 0.250 ± 0.015 | 3489 ± 382 | 1207 ± 90 | 4.82 ± 0.46 | 0.291 ± 0.010 | 1.875 ± 0.035 | 14.58 ± 0.32 | 11.3 ± 2.2 | 64.5 |
| LUM4775 | MAR-30 | | 0.239 ± 0.015 | 2454 ± 337 | 1046 ± 107 | 4.38 ± 0.52 | 0.345 ± 0.013 | 1.804 ± 0.014 | 13.94 ± 0.13 | 11.3 ± 1.7 | 59.5 |

Ḋ is the dose rate. $D_e$ is the equivalent dose. $D_0$ is the characteristic saturation dose of the electron trap. n/N is the signal saturation level, which is the ratio of the measured TL signal intensity and the maximum TL signal intensity at saturation. $T_C$ is the effective closure temperature calculated from the cooling rate, the surface temperature and the apparent TL age. Cooling rates of the fault clasts have not been modeled as their TL ages indicate a signal resetting event at ~2.5 Ma, rather than cooling ages. Errors for all values are 1σ.
ᵃThe 100–200 μm grain size fraction was used for all samples. However, a grain size fraction of 63–100 μm was also measured for sample LUM4525, to make comparison with the 100–200 μm grain size fraction.

signal of a sample from a greater depth is more easily to be fully reset due to the higher stress[79]. At ~2.5 Ma, the clasts were buried at depths below 1 km, which would promote the complete resetting of the TL signal during faulting. Another independent evidence supporting the complete resetting scenario is that the dolomite clasts were sampled from an exhumed fault damage zone along the basin-boundary faults of the AIB and their TL ages are compatible with the inception of syn-rift sedimentation in the area[46]. With a cooling rate of 11.3 °C Ma⁻¹, a TL age of 2.5 Ma corresponds to a geothermal temperature of 38 °C, under which the thermal detrapping/trapping rate ratio of the TL signal is smaller than 0.05 (Fig. 4c). Additionally, the TL signal's thermal lifetimes at 35 °C of the clasts are calculated to be 28–99 Ma, which are already more than 10 times longer than the 2.5 Ma age. The thermal lifetimes at lower temperatures are even longer. Thus, the TL signal can be regarded as sufficiently stable at temperatures under 35 °C, and thermal loss is negligible. Consequently, the apparent TL ages of ~2.5 Ma can represent the true age when the resetting event happened. Given these considerations, we hypothesize that the TL signals of the fault clasts were reset completely during a major seismic event (or a major seismic phase with multiple seismic events) at ~2.5 Ma ago, likely associated with the onset of $D_2$ post-orogenic extensional faulting in the central Apennines. However, based on the field evidence documenting in situ rock shattering associated with the MMF development, which suggests that fluid- and/or gas-mediated fracturing likely contributed to the development of the fault damage zone[58,82], we cannot exclude a contribution given by the structurally-controlled fluid-rock interaction in resetting the TL signals in the damage zone of the MMF.

**Contribution to refining the central Apennine neotectonics**

The Early Pliocene (4.60 ± 0.35 Ma) exhumation age of the carbonate bedrock at the northeastern shoulder of the AIB indicates an exhumation event that predated the onset of post-orogenic extension in the central Apennines. Assuming a geothermal gradient of 25 °C km⁻¹[83] and a $T_S$ of 10 °C, the $T_C$ of ca. 60 °C corresponds to an exhumation depth of ca. 2.0 km (Fig. 4c). From the cooling rates of the two bedrock samples (11.3 ± 2.2 and 11.3 ± 1.7 °C Ma⁻¹), the exhumation rates are calculated to be 0.45 ± 0.09 and 0.45 ± 0.07 mm yr⁻¹, respectively. The mean exhumation rate is 0.45 ± 0.06 mm yr⁻¹. It indicates that the carbonate bedrock traveled through a paleo-geothermal temperature of 60 °C (2 km depth) at 4.60 ± 0.35 Ma and continued to be exhumed to the surface with an average exhumation rate of 0.45 ± 0.06 mm yr⁻¹. Notably, these Early Pliocene TL ages are in principle compatible with timing of the orogenic thrusting in the region (late Messinian-Early Pliocene in age[33]). Consequently, the bedrock dolomite TL dating provided in this study suggests that exhumation of the basin shoulders of the AIB was likely primarily driven by thrust-related erosion during the $D_1$ nappe stacking and orogenic construction (Fig. 5a).

The Early Pleistocene TL ages between 2.4 and 2.7 Ma (average: 2.53 ± 0.13 Ma) as derived from the fault clasts within the damage zone of the MMF indicate a fault-related signal resetting event, which can be tentatively interpreted as the timing of the onset of the $D_2$ post-orogenic extensional faulting in the central Apennines. At regionals scale, this timing is in line with (i) the major erosion/exhumation event at 3.0–2.5 Ma during regional uplift[49,50,52,84–86]; (ii) the stratigraphic age of the basal syn-rift deposits across the backbone of the orogenic chain[46]; and (iii) the direct dating of extensional faulting in the central Apennines[48] (Fig. 5b). After the signal resetting at 2.53 ± 0.13 Ma, the dolomite TL signals of the fault clasts within the damage zone of the MMF started to accumulate again as induced by the progressive hanging wall downfaulting during formation of the AIB shoulder. In this scenario, assuming (i) the maximum ~1.8 km stratigraphic separation across the MMF[32], and (ii) an average fault dip angle of 65° for the extensional MMF, a maximum long-term (tectonic- and erosion-related) exhumation rate of 0.71 ± 0.04 mm yr⁻¹ can be thus derived for the footwall carbonate bedrocks, converting to an average maximum fault slip rate of 0.78 ± 0.04 mm yr⁻¹ (Fig. 5c). This value is in principle compatible with both the short-term (paleoseismic) and long-term estimates of fault slip rates in

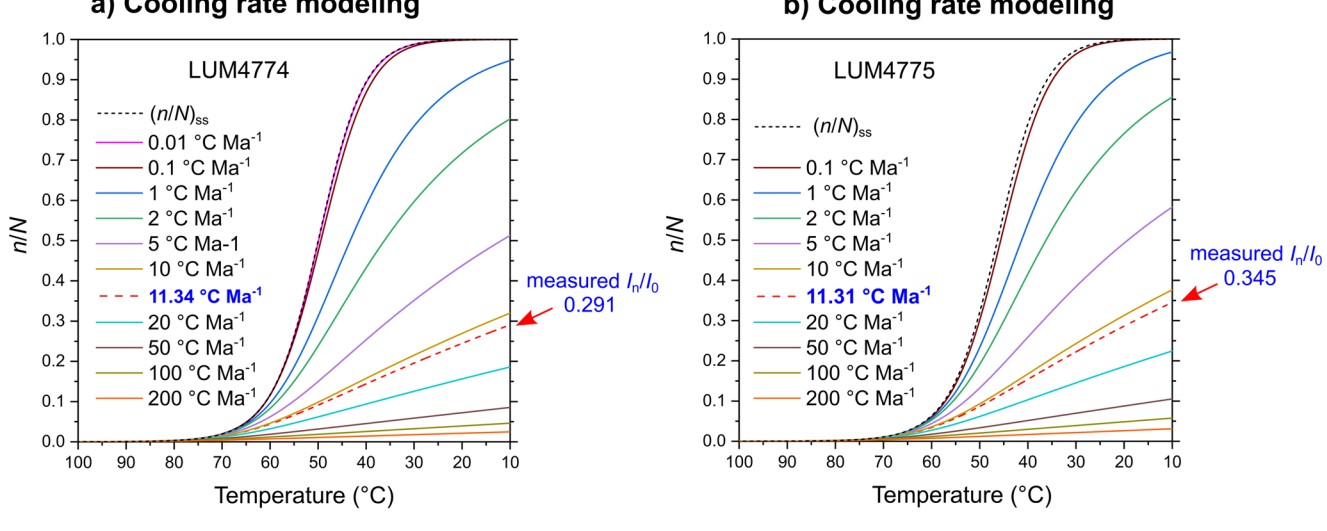

### c) Detrap/trap ratio with cooling

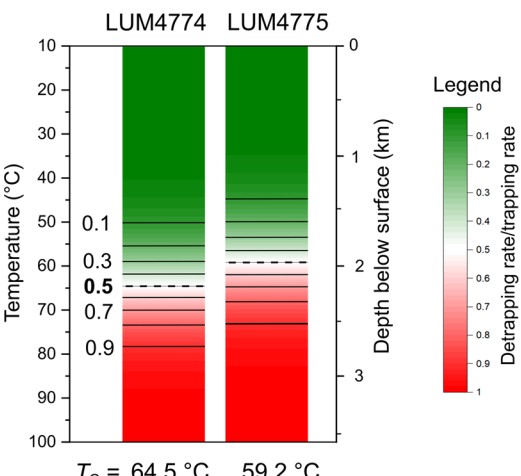

### d) $T_C$ with cooling rate and dose rate

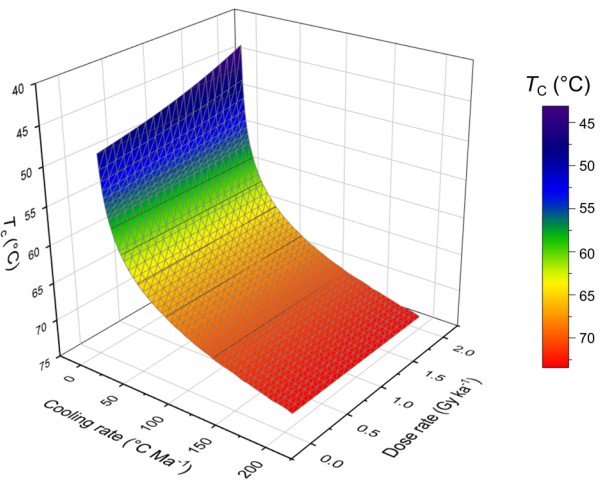

**Fig. 4 | Cooling rate modeling and closure temperature estimation. a** TL signal (signal saturation level, $n/N$) growth with temperature under different cooling rates for the bedrock sample LUM4774. With a cooling rate of 11.3 °C Ma$^{-1}$, the modeled signal saturation level equals to the measured one. The $(n/N)_{ss}$ means the signal at the thermal equilibrium state under a certain temperature, i.e., the sample stays at that temperature for infinitely long time. **b** TL signal growth with temperature under different cooling rates for the bedrock sample LUM4775. **c** Detrapping/trapping rate ratio *vs* Temperature (Depth) for the two bedrock samples. Depth below surface is calculated based on a geothermal gradient of 25 °C km$^{-1}$. **d** Modeling the effective closure temperature ($T_C$) under different cooling rates (2–10 °C Ma$^{-1}$ with an interval of 1 °C Ma$^{-1}$, and 10–200 °C Ma$^{-1}$ with an interval of 5 °C Ma$^{-1}$) and dose rates (0.1–2.0 Gy ka$^{-1}$ with an interval of 0.1 Gy ka$^{-1}$). For the $T_C$ modeling, the $E$ is fixed at 1.80 eV, the $s$ at $10^{14}$ s$^{-1}$, the $D_0$ at 5000 Gy, and the surface temperature at 10 °C.

the central Apennines, which are in the order of 0.4–1.2 mm yr$^{-1}$ (preferred 0.6–0.8 mm yr$^{-1}$)$^{136}$ and 0.9 mm yr$^{-1}$$^{148}$, respectively. Additionally, our long-term exhumation rate of 0.71 ± 0.04 mm yr$^{-1}$ aligns with the short-term tectonic throw rates of 0.4–0.6 mm yr$^{-1}$ (cumulative ~1.0 mm yr$^{-1}$) observed across different MMF segments over the last 35 kyr$^{81}$. This evidence suggests that, over the long-term, tectonic rates may have remained nearly constant in the region since Early Pleistocene. It also highlights the potential of dolomite TL dating for assessing mean spatio-temporal fault slip rates, contributing to a better understanding of the long-term tectonic evolution in seismically active regions.

### The potential of the dolomite TL thermochronometry

Due to the low $D_0$ values, quartz and feldspar luminescence thermochronometry tools are only applicable in rapidly exhuming terrains with cooling rates higher than 200 °C Ma$^{-18}$, and the quartz ESR thermochronometry is only applicable for cooling rates higher than 20–50 °C Ma$^{-1}$$^{12,13}$. For the six dolomite samples in this study, the $D_0$ values of the 350 °C TL peak

range between ~2500 and 5900 Gy (Table 1). Empirically, taking a $2D_0$ as the maximum value below which the $D_e$ can be measured with sufficient accuracy$^9$, the maximum measurable $D_e$ would be in the range of 5000–11800 Gy. The dose rates range from 0.24 to 1.02 Gy ka$^{-1}$. Thus, the corresponded maximum TL ages are 11–28 Ma, which is 1–2 orders of magnitude higher than the dating limits of quartz and feldspar. The applicable cooling rate range of each sample can be modeled by an $n/N$ range of 5–85%. For an $n/N$ value smaller than 5%, the cooling rate is too high and only a small amount of the natural signal has accumulated for the sample exhumed to the surface, which will result in large uncertainty in signal measurements. For an $n/N$ value higher than 85%, the cooling rate is too low, and the natural signal is too close to saturation level. Based on the parameters ($E, s, D_0, \dot{D}$) of the six dolomite samples in this study, the accessible minimum cooling rate is 1.7–4.1 °C Ma$^{-1}$ and the accessible maximum cooling rate is 92–232 °C Ma$^{-1}$ (Fig. S6).

The TL signals of dolomite samples in this study show no signs of anomalous fading (details in the Methods section), which makes the cooling history modeling simpler compared to the feldspar luminescence

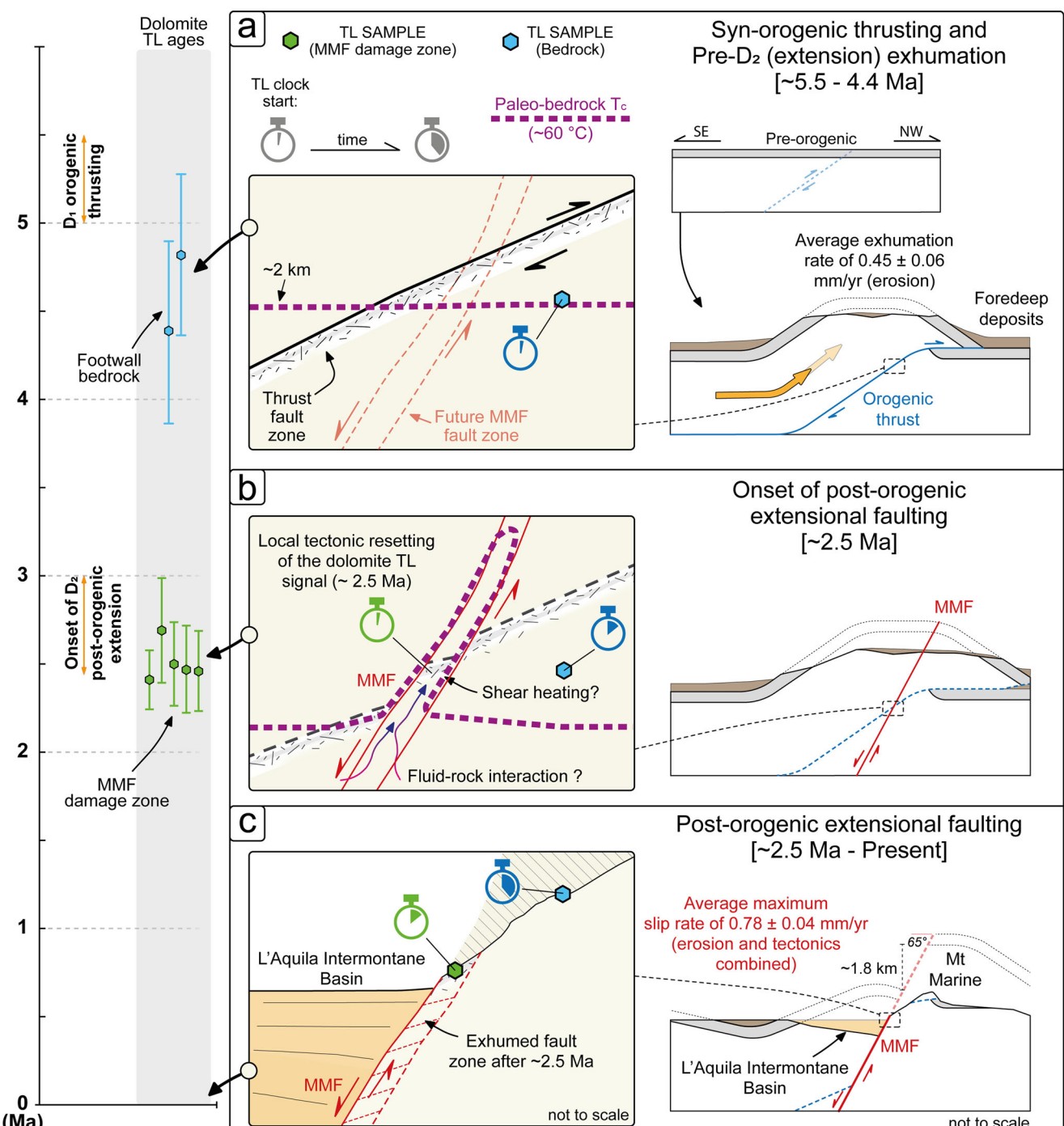

**Fig. 5 | Tectonic and exhumation history of the northeastern shoulder of the L'Aquila Intermontane Basin.** Left panel: Summary of the dolomite TL ages framed within the available tectono-stratigraphic temporal constraints for the $D_1$ syn-orogenic thrusting and $D_2$ post-orogenic extension stages in the central Apennines. Right panel: Schematic exhumation and tectonic history at the footwall of the Monte Marine Fault (MMF) along the northeastern shoulder of the L'Aquila Intermontane Basin as reconstructed from the dolomite TL ages. (**a**) Syn-orogenic exhumation of the carbonate bedrock; (**b**) Possible resetting of the dolomite TL signal along the nascent MMF zone; (**c**) Final exhumation of the MMF zone caused by the westward propagation of the extensional faulting and opening of the L'Aquila Intermontane Basin.

thermochronometry[87,88]. Furthermore, it avoids the additional uncertainty associated with the measurements of fading rates. The $E$ values of the ~280 °C TL are in the range of 1.52–1.60 eV, smaller than those of the 350 °C TL peak (1.71–1.88 eV) (Table S3). The lower thermal stability of the ~280 °C TL peak results in a smaller $T_C$ compared to the 350 °C TL peak, thus giving it the potential to constrain the late-stage cooling history. However, the low thermal stability also means that the thermal loss of the 280 °C TL peak may still be significant at the surface temperature (e.g.,

10 °C). A thermal equilibrium state of the signal, $(n/N)_{ss}$, will be reached for a sample which stays infinitely long at the surface. The $(n/N)_{ss}$ is dependent on the surface temperature, and can be calculated from Eq. (2) by setting $d\left(\frac{n}{N}/dt\right)$ to zero:

$$(n/N)_{ss} = \frac{\dot{D}/D_0}{\dot{D}/D_0 + 1/\tau} \tag{5}$$

For the 280 °C TL peak, the thermal lifetime at 10 °C ($\tau_{10}$) of the six samples is in the range of 1.03–7.56 Ma (Table S3), making their $(n/N)_{ss}$ values within 0.15–0.72 (Fig. S7a, b). While these low signal saturation levels at the thermal equilibrium state can be used for surface paleothermometry[21,22,89], it is a disadvantage for thermochronometry. Taking LUM4774 as an example, an $(n/N)_{ss}$ value of 0.15 means that all samples with different cooling rates always have signal saturation levels lower than 0.15 at a surface temperature of 10 °C (Fig. S7b). In this case, the uncertainty in $I_n/I_0$ measurements will result in a very high uncertainty in the modeled cooling rate. Thus, these low $(n/N)_{ss}$ values hinder the application of thermochronometry. Instead, the high thermal stability of the 350 °C TL peak makes the $1/\tau << \dot{D}/D_0$, resulting in $(n/N)_{ss}$ values very close to 1 (Fig. 4a, b and S7c, d). The TL curve deconvolution shows that the ~280 °C TL peak actually contains two peaks at 250 °C and 295 °C, respectively (Fig. S3). If the thermal kinetic parameters and growth curve parameters can be determined individually for these two TL peaks, the 295 °C TL peak should have relatively higher thermal stability compared to the combined 280 °C TL peak, and it might be suitable to be used together with the 350 °C TL peak for thermochronometry, to study the cooling history in different temperature regimes.

The cooling history modeling was performed by assuming that one single trap with a unique $E$ is responsible for the 350 °C TL peak of dolomite. However, there is also a possibility that the 350 °C TL peak corresponds to a group of electron traps with their $E$ values in a Gaussian distribution, which is the case for quartz ESR signal[12–14,69]. The reported standard deviation values of $E$ ($\sigma E$) for trapped charge thermochronometry are almost always smaller than 0.2 eV[12–14,69]. We performed cooling history modeling for LUM4774 assuming Gaussian distribution of $E$ values, with the $\sigma E$ value ranging from 0 to 0.3 eV (Fig. S8a). The modeled cooling rate decreases slightly with higher $\sigma E$ values, and reaches 10.3 ± 2.0 °C Ma$^{-1}$ with a $\sigma E$ of 0.2 eV, which is still consistent (considering the uncertainty) with the cooling rate of 11.3 ± 2.2 °C Ma$^{-1}$ based on the single-trap modeling ($\sigma E$ = 0) (Fig. S8b). We also simulated the signal growth under different cooling rates with a fixed $\sigma E$ of 0.2 eV. With the Gaussian distribution of traps, the luminescence signal starts to accumulate at higher temperatures compared to the single-trap scenario, due to the contribution of deeper traps; meanwhile, the $(n/N)_{ss}$ value at the surface temperature becomes lower than 1, due to the contribution of shallower traps (Fig. S8c, d). Regarding the $n/N$ values at the surface temperature, significant deviation between the two scenarios starts to occur only when the cooling rate is below 10 °C Ma$^{-1}$.

To conclude, the dolomite TL thermochronometry significantly broadens the applicable cooling rate ranges for the trapped charge thermochronometry compared to quartz and feldspar. Moreover, in carbonate bedrocks, where traditional thermochronometry is not applicable due to the absence of target minerals (e.g., zircon, apatite, quartz, feldspar), dolomite thermochronometry provides a viable and effective approach for reconstructing the cooling and exhumation histories, to contribute to refining the long-term tectonic evolution at a regional scale. Finally, our study also indicates that the TL signal of dolomite in the fault zone can be completely reset during seismic events, and the corresponding TL age can be used to constrain the paleo-seismic activity along major fault strands.

## Methods
### Sample information
Four dolomite clasts (LUM4524, LUM4525, LUM4771, LUM4773) at different locations in the fault damage zone of the MMF were collected in June 2022, for TL dating. The clasts are of sphere shape, with diameters of ca. 3–5 cm, and weights of ca. 40–200 g. Five samples from the cataclasite matrix were collected close to these dolomite clasts. Two dolomitized bedrock samples were also collected for TL dating at different distances from the fault damage zone (~20 m for LUM4775 and ~100 m for LUM4774, respectively). Two fault clast samples, two matrix samples and two dolomitized bedrock samples were tested with XRD analyses with a PANalytical MPD Pro XRD equipment at the Federal Institute for Geosciences and Natural Resources (BGR), Hannover, Germany, and the results indicate that

all the samples are dominated by dolomite mineral with trace amount of calcite (Fig. S1). Details of sample information and adopted analytical methods for the samples are provided in Table S1.

### U, Th, K and REY measurements
The mass fractions of rare earth elements and the pseudolanthanoid Y, as well as U, Th, and K (required for dose rate calculations) were measured after wet chemistry with a QQQ-ICP-MS instrument in the Laboratory of the Geochemistry Research Unit at BGR. The samples were ground into powder in an agate mortar. 250 mg of the dried powders were digested by 10 ml of 5 M $HNO_3$ (suprapure grade) inside acid-cleaned Savillex PTFE beakers. The suspensions inside the beakers were heated to 70 °C on a hotplate for 2 h. Afterwards, the suspensions were cooled down to room temperature, and were filtered with acid/DI-cleaned filter syringes with a pore size of 0.2 μm. The filtered solutions were diluted with DI to achieve a final molarity of 1 M $HNO_3$. These solutions were measured with a Thermo Fisher iCAP TQ ICP-MS. The REY, Th and U were measured in KED mode with He as reaction gas. Potassium was measured on mass 39 with $O_2$ as reaction gas to minimize interferences by e.g., $^{38}Ar^1H$. The European Shale dataset (EUS[90]) was used for normalization of the REY data. The reference material J-Do1 (dolostone; issued by the Geological Survey of Japan) was digested along with the samples for quality control. The measured K mass fraction of J-Do1 was 18.8 ppm (reference: 19.3 ppm; georem Database; accessed on 12.05.2023), and the measured Th and U mass fractions of J-Do1 were 0.044 and 0.84 ppm (published literature values are <0.04 and 0.88 ppm[91]). The measured mass fractions of the rare earth elements and Y in JDo-1 deviated between +2 and +10% in comparison to published JDo-1 data[91].

### Sample preparation for TL measurements
Samples used for TL dating were prepared under subdued red light in the luminescence laboratory in Leibniz Institute for Applied Geophysics (LIAG), Hannover, Germany. Artificial bleaching tests show that the natural TL signals of dolomite samples can be depleted by ~50% after being exposed to a solar simulator (model: UVACUBE 400 of Dr. Hönle UV Technology) for 2 h (Fig. S9). To remove the effect of sunlight exposure, the clast samples were firstly immersed under the 32% HCl acid for reaction. The volume of the HCl acid used of each clast was calculated to dissolve about half weight of the clast. Usually, after 1–2 h, the reaction became very weak. The clasts were taken out and washed by distilled water and weighed. The weight loss was generally 40%. The two bedrock samples were large pieces of rocks. They were firstly crushed into small fragments under the subdued red light. The fragments from the middle part of the bedrock were chosen to be further reacted with the 32% HCl acid to remove ~40% of the weight. Afterwards, these samples were gently crushed in a steel mortar under water. The products were wet sieved and the 100–200 μm fraction was separated for $D_e$ measurements. For one sample (LUM4525), the 63–100 μm fraction was also separated and its $D_e$ was measured for comparison. The dolomite grains were mounted on stainless steel disks to prepare the aliquots for TL measurements.

### TL emission spectra measurements
The TL spectra were measured using an Andor Shamrock 163 Czerny-Turner spectrograph coupled to an Andor Newton DU920P back-illuminated charge-coupled device (CCD) camera built into a Lexsyg Research reader at the Institute of Geography, Justus Liebig University of Giessen. A diffraction grating with a groove density of 300 lines mm$^{-1}$ and a blaze wavelength of 500 nm was used. For wavelength calibration, we used the ionization emission lines of mercury and REY from a 'white-light' ceiling lamp[92]. Spectra were efficiency-corrected using a spectral-response function obtained from the product of efficiency curves provided by the manufacturers, including for a 3-mm thick SCHOTT KG-3 filter, diffraction grating, CCD camera, and fiber optic bundle. Wavelengths spanning 350–650 nm were transmitted with over 50% efficiency. A background spectrum obtained at room temperature was subtracted from the

measurements prior to any data processing. The spectra contained abrupt signal spikes probably caused by cosmic rays. These outliers were removed by applying a running median of length 3 along the temperature axis and then along the wavelength axis iteratively 6 times via the 'apply_CosmicRayRemoval' function of the 'Luminescence' (v0.9.19) R package[93].

### $D_e$ measurements

Luminescence measurements were performed with a Risø TL/OSL reader 15. A 2-mm thick SCHOTT BG-39 filter (transmission window of 350–600 nm) was placed in front of an EMI 9235QB photomultiplier tube, to minimize the blackbody emission signal. The Risø reader has an attached $^{90}Sr/^{90}Y$ beta source. The dose rate was calibrated to be $0.121\,Gy\,s^{-1}$ for coarse grains on stainless steel disks in April 2021, and corrected monthly for the decay of the source. The multiple-aliquot additive-dose (MAAD) protocol were used to measure the $D_e$ (Table S2). The TL measurements were always under $N_2$ flow, and a heating rate of $5\,°C\,s^{-1}$ was used. The aliquots were preheated to 260 °C to remove the unstable low-temperature TL signals. Then, they were heated to 450 °C (with background subtraction) to record the TL signals of interest, i.e., the ~280 and 350 °C TL peaks. To normalize the inter-aliquot variation due to different number of grains on each disk, a small dose (5 Gy) was given to the aliquots and the corresponded TL signal was measured up to 200 °C. The 80–180 °C signal of this small dose was used for inter-aliquot normalization for the ~280 and 350 °C TL peaks, which is equivalent to mass normalization[94].

As the two TL peaks observed in the natural signals (peak I centered at ~280 °C and peak II at ~350 °C, respectively) have different thermal stabilities and thus different closure temperatures, theoretically they can record cooling history information at different temperature regimes[6,7,95,96]. For the laboratory dosed TL signal (regenerative TL), peak I of ~280 °C cannot be identified, and there appears to be three TL peaks at ~150, 250, and 350 °C, respectively (Figs. 3b and S3). However, peak deconvolution shows that the regenerative TL curves cannot be fitted successfully with these three peaks (Fig. S3a, b). Instead, they can be fitted perfectly with four peaks centered at ~150, 250, 295, and 350 °C, respectively (Fig, S3c, d). The peak at ~350 °C of the regenerative TL curve is same as the peak II in the natural TL curve. The lowest TL peak at 150 °C has very short lifetime and thus is absent in the natural TL signal. The 250 °C and 295 °C TL peaks are located close to each other and the 250 °C TL peak has a much higher intensity than the 295 °C TL peak. Consequently, these two peaks cannot be distinguished from each other in the regenerative TL curve, which only shows one peak centered at 250 °C, with a wide shoulder on the right side. Because the 250 °C TL peak has a low thermal stability on the geological timescale, its intensity is low in the natural TL curve. Thus, the peak I (at ~280 °C) in the natural TL curve is closer to 295 °C rather than 250 °C. For $D_e$ estimation, a preheat to 260 °C was performed before the laboratory dosed TL signal measurements, to remove the unstable low temperature TL signals. With this heat treatment, the peak I of the regenerative TL curve is centered at 275–280 °C. Though the position of peak I of the regenerative TL signal is close to the peak I of the natural signal, age determination based on the synthetic peak I is still not meaningful, as it contains components from two peaks (two electron traps) with different thermal lifetime parameters and saturation characteristics. Since the thermal kinetic parameters and dose response curve information for each of the two peaks centered at 250 °C and 295 °C cannot be accurately determined due to the proximity of these two peaks, the cooling history modeling of the synthetic peak one is not meaningful. Thus, we only discuss the TL ages and cooling rate modeling results of the 350 °C TL peak.

### Thermal lifetime parameter measurements

The peak shifting method[97] was applied to estimate the thermal lifetime parameters for the dolomite samples, i.e., the activation energy/trap depth ($E$) and frequency factor ($s$). The aliquots were repeatedly administered a fixed dose of ~500 Gy, preheated to 260 °C to remove the low-temperature signal (with a heating rate of $5\,°C\,s^{-1}$), and then heated to 450 °C with different heating rates to record the TL signal. The heating rates used for peak shifting were 0.1, 0.2, 0.5, 1.0, and $2.0\,°C\,s^{-1}$. These relatively low

heating rates can minimize the thermal lag between the heating plate and the sample grains. The peak positions of the ~280 °C and 350 °C TL peaks shifted to lower temperatures with smaller heating rates, and thus the $E$ and $s$ values can be deduced[97] (Fig. S10). The $E$, $s$ values and the lifetimes at 10 °C and 20 C are summarized in Table S3. Note that the $E$ and $s$ values for the ~280 °C TL peak are not meaningful as the ~280 °C TL peak is a combination of two peaks.

### Anomalous fading tests

Anomalous fading is a phenomenon that the luminescence signal decays at room temperature although the signal is thermally table[98]. Studies on TL signals of dolomite samples from the Ataka Mountain in Egypt indicate the existence of fading[99,100]. It is not clear whether anomalous fading is ubiquitous in dolomite samples or not. Here we performed fading tests on one clast sample LUM4524 and one bedrock sample LUM4774. A multiple-aliquot test was performed on LUM4524. Thirteen aliquots of LUM4524 were prepared and heated to 450 °C to remove the natural signals. Then, these aliquots were given doses of 1200 Gy, preheated to 260 °C, and divided into two groups. For one group, their TL curves were measured up to 450 °C immediately after the preheat treatment. For the other group, the aliquots were stored in darkness for 14 days after the preheat treatment, before their TL curves were measured. The intensity of the TL signals after a storage of 14 days shows no depletion compared to the TL signals measured promptly, indicating that there is no fading (Fig. S11a). It is noteworthy that the TL curves measured after a 14-day delay shifted by several degree Celsius compared to the TL curves measured immediately. The reason is not clear yet, but it might be related to the poor reproducibility of the luminescence reader during the two measurements.

The single-aliquot regenerative-dose (SAR) protocol was also applied to measure the fading rate ($g$-value) of samples LUM4524 and LUM4774 (3–4 aliquots for each)[101]. The regenerative dose and test dose were fixed at 600 Gy. The sensitivity corrected signals of the 350 °C TL peak were measured after different delay times (from 0.62 h to 226 h). The mean $g$-values ($t_c = 0.62\,h$) are $0.57 \pm 0.42$ and $0.26 \pm 0.21$ %/decade for LUM4524 and LUM4774, respectively (Fig. S11b, c). These near-zero $g$-values also indicate no fading for the dolomite samples in this study.

### Dose rate calculation

The samples are almost pure dolomite from the XRD analyses, and thus an infinite homogeneous medium can be assumed for dose rate calculation. The almost identical $D_e$ values between the 63–100 µm and 100–200 µm fractions of LUM4525 (Table 1) further proves that the initial grain sizes of the dolomite crystals have no influence on the dose rate calculation. Within a homogeneous medium, the alpha dose rate contributes most to the total dose rate, making the alpha efficiency a key parameter[102]. In this study, we apply the $S_a$ value system[103] to describe the alpha efficiency. With the $S_a$ value, the alpha flux can be directly converted into the effective alpha dose rate, and the $S_a$ value system is less sensitive to the energy of alpha particles compared to the $k$-value system. $S_a$ values of two dolomite clast samples (LUM4524, LUM4525) were measured in the luminescence laboratory in University Bordeaux Montaigne. The samples were ground into fine power and the 4–11 µm fraction was separated following the Stokes law. Aliquots were prepared with steel cups from the Lexsyg instruments. These cups have an internal diameter of 8 mm. About 0.6 g of fine grains were deposited onto steel cups, making the covered cup surface thin-layered with a surface density of 1 mg $cm^{-2}$[104]. The $S_a$ value measurements were performed with a Lexsyg smart TL/OSL reader (always under $N_2$ flow), equipped with a $^{90}Sr/^{90}Y$ beta source which had a dose rate of $0.133\,Gy\,s^{-1}$ (in March 2023) for fine grains (4–11 µm) on the steel cups. Because the samples were ground into fine powder, strong spurious TL signals appeared at the high temperature range, similar to what was observed in calcite[105]. The aliquots were firstly heated to 450 °C for four times to remove any natural and spurious TL signals. Afterwards, the aliquots were administered a beta dose of 100 Gy and heated to 450 °C. This dosing and heating step was repeated for two more times to stabilize the sensitivity change of the aliquots. The stabilized

aliquots were moved out of the Lexsyg smart reader and irradiated by a home-made $^{241}$Am alpha source for 12 h. The $^{241}$Am alpha source had an alpha flux rate of $1.9 \times 10^5$ α-particles s$^{-1}$ cm$^{-2}$[104]. The equivalent beta doses of these aliquots were measured using the Lexsyg smart reader. Since the aliquots were already stabilized, the sensitivity change is stable and the SAR TL protocol (Table S4) can be used for the beta $D_e$ measurements[102,106]. Note that for $S_a$ value measurements, the heating rate was set to be 1 °C s$^{-1}$. With this lower heating rate, the peak positions of the ~280 °C and ~350 °C TL peaks with a heating rate of 5 °C s$^{-1}$ shifted to ~245 and 320 °C, respectively (Fig. S12). The beta $D_e$ was divided by the received alpha flux to calculate the $S_a$ value. The $S_a$ values for the 350 °C TL peak are $27.6 \pm 2.7$ and $26.7 \pm 2.0$ μGy/(1000α*cm$^{-2}$) for sample LUM4524 and LUM4525, respectively. The mean $S_a$ value of $27.2 \pm 1.7$ μGy/(1000α*cm$^{-2}$) was used for another two dolomite clast samples (LUM4771 and LUM4773) for alpha dose rate calculation. The $S_a$ values of the two dolomitized bedrock samples (LUM4774 and LUM4775) were measured in the Institute of Geography, Justus Liebig University of Giessen, with a Lexsyg Research reader which had an attached $^{241}$Am alpha source. Since the alpha flux of the $^{241}$Am alpha source in Giessen has not been calibrated, the dolomite clast samples LUM4524 and LUM4525 were repeatedly measured in Giessen and their $S_a$ values were applied to calibrate the $S_a$ values of the two bedrock samples. The $S_a$ values of the LUM4774 and LUM4775 are much smaller compared to the clast samples, which are $15.9 \pm 1.6$ and $15.7 \pm 1.7$ μGy/(1000α*cm$^{-2}$), respectively. The ranges of alpha particles[107] in dolomite ($\rho = 2.85$ g cm$^{-3}$) were obtained from the software of 'The Stopping and Range of Ions in Matter' (SRIM-2013). With these alpha ranges, alpha flux of 1 ppm U and 1 ppm Th in dolomite were calculated to be 18013 and 5047 cm$^{-2}$*year$^{-1}$, respectively. Alpha dose rate was calculated by converting the alpha flux with the $S_a$ value. Since the alpha particle energy spectrum of the home-made alpha source in the University Bordeaux Montaigne is different from the alpha spectrum in nature, a correction factor of 0.92 for U and a correction factor of 0.96 for Th were used for the $S_a$ value when calculating the alpha dose rate[102]. For beta and gamma dose rate calculation, the conversion factors of Guérin et al.[108] were applied. Water contents were taken as zero. Cosmic ray dose rate decays exponentially with depth[109], and it is lower than 0.01 Gy ka$^{-1}$ once the depth is larger than 35 m (Fig. S13). Since the samples were more than 35 m below the surface for most time during the exhumation history, the cosmic ray is not taken into consideration for environmental dose rate calculation. The details of dose rates are listed in Table S5.

### Cooling rate modeling

The modeling of cooling rate is performed with home-made scripts written in R[110]. We modeled the trapped charge accumulation with different cooling rates, from an initial temperature of 100 °C to a surface temperature of 10 °C, and then compared the modeled $n/N$ with the measured one, until the best match was found. The modeling was performed at a time interval of 1 kyr. Note that at the temperature of 100 °C, the lifetime of the trap is very short, thus the electron detrapping rate is much higher than the trapping rate, and the system can be regarded as a totally open system. The error of the cooling rate is estimated with the Monte Carlo method, by repeating the cooling rate modeling 500 times with Gaussian distributions of $T_S$, $n/N$, $\dot{D}$, $D_0$, $E$ and log($s$). An error of 1 °C has been assigned to the $T_S$. Since the 1σ error is applied for generating the Gaussian distributions of the $T_S$, $n/N$, $\dot{D}$, $D_0$, $E$ and log($s$), the standard deviation of the simulated 500 cooling rates corresponds to 1σ error as well.

For cooling rate modeling with Gaussian distribution of electron traps, a group of $E$ values were generated within the range of 1.0–3.0 eV, by an interval of 0.01 eV. The probability of each $E$ value is calculated following the Gaussian function, based on the mean $E$ and $\sigma E$. The probabilities were normalized to make sure the sum of the probabilities is 1. The initial temperature for cooling history modeling was set to 150 °C instead of 100 °C. Signal growth paths were modeled for each $E$ value and the weighted sum (the weight is the probability of each $E$) of these individual signal growth paths represent the overall natural signal growth path.

## Data availability

The thermoluminescence data and the cooling rate modeling results are stored in Figshare repository[111] (https://doi.org/10.6084/m9.figshare.28462943.v1).

## Code availability

The R scripts used for cooling history modeling are available in Figshare repository[111] (https://doi.org/10.6084/m9.figshare.28462943.v1).

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

## Acknowledgements

The authors are grateful to Nobert Mercier and Chantal Tribolo from the University Bordeaux Montaigne, Sebastian Kreutzer from the Heidelberg University, Markus Fuchs and Thomas Kolb from the Justus Liebig University of Giessen, for their help and discussion about alpha efficiency measurements. The authors would like to thank Kristian Ufer and Niko Götze from BGR for their help in XRD measurements. The comments from Christoph Glotzbach from the University of Tübingen on an earlier version of the manuscript are appreciated. The authors especially thank Pierre Valla and the other two anonymous reviewers whose insightful comments have greatly improved our manuscript. The authors declare that no permissions were required for sample collection in the studied area. This work has been partially supported by the European Research Council under Horizon 2021 research and innovation program (ERC-2021-COG, Grant No. 101045217) awarded to Faysal Bibi (and Sumiko Tsukamoto as a collaborator). G.A., P.C., D.C., and F.R. acknowledge support provided by the grant MIUR-Italy Dipartimenti di Eccellenza (ARTICOLO 1, COMMI 314 – 337 LEGGE 232/2016) to the Department of Science, Roma Tre University.

## Author contributions

J.Z. conceived the study, collected the samples, performed luminescence measurement and cooling rate modeling, and wrote the original draft. G.A. collected the samples, wrote the original draft. F.R. conceived the study, collected the samples, and wrote the original draft. V.A. collected the samples, reviewed the manuscript. D.K. performed REY elements analysis and data interpretation, wrote the related method section, and reviewed the manuscript. M.S.G. made the TL emission spectra measurements, wrote the related method section, and reviewed the manuscript. D.C. collected the samples, reviewed the manuscript. P.C. collected the samples, reviewed the manuscript. S.T. conceived the study, reviewed the manuscript. All authors contribute to data interpretation.

## Funding

## Competing interests

The authors declare no competing interests.
