## [Transparent Peer Review file · Communications Earth & Environment]

Dolomite luminescence thermochronometry reconstructs the low-temperature exhumation history of carbonate rocks (central Apennines, Italy)

Corresponding Author: Dr Junjie Zhang

Version 0:

Decision Letter:

Dear Dr Zhang,

Your manuscript titled "Dolomite thermochronometry applied in central Apennines: a novel tool for tectonic reconstruction in carbonate rocks" has now been seen by 3 reviewers, whose comments are appended below. You will see that they all consider the thermoluminescence (TL) dating on dolomite as a new promising tool for carbonate thermochronology. However, they have raised quite substantial concerns that must be addressed. In light of these comments, we cannot accept the manuscript for publication, but would be interested in considering a revised version that fully addresses these concerns. Please, carefully consider the following editorial threshold:

- 1) Provide firm and sufficient evidence to demonstrate that dolomite thermochronometry is a robust method to reconstruct the exhumation and cooling histories of carbonate rocks.
- 2) Include alternative hypotheses for your data interpretation and consider points raised by Revs #2 and #3. In particular take care of Rev #3's note concerning the regional uplift of the Apennines and the possibility that the older ages are related to it. The age of the clasts from the fault damage zone can be provoked by direct fault activity (fluid circulation, re-crystallization or shear heating) as suggested by Rev # 2.
- 3) Include in the main text some information about the technique to better permit to the readers to follow the results and quality of data.

We hope you will find the reviewers' comments useful as you decide how to proceed. Should additional work allow you to address these criticisms, we would be happy to look at a substantially revised manuscript. If you choose to take up this option, please either highlight all changes in the manuscript text file, or provide a list of the changes to the manuscript with your responses to the reviewers.

When resubmitting, please provide a point-by-point response to the reviewers' comments. Please submit your responses as a separate file, distinct from your cover letter where you can add responses to the Editors' comments that you do not want to be made available to the reviewers. Word files are preferred. We recommend that any figures, tables or graphs that are included in the response to reviewers are also included in the main article or Supplementary Information.

If the revision process takes significantly longer than three months, we will be happy to reconsider your paper at a later date, as long as nothing similar has been accepted for publication at Communications Earth & Environment or published elsewhere in the meantime.

Please use the following link to submit your revised manuscript, point-by-point response to the reviewers' comments with a list of your changes to the manuscript text (which should be in a separate document to any cover letter), a tracked-changes version of the manuscript (as a PDF file) and any completed checklist:

Link Redacted

Please do not hesitate to contact us if you have any questions or would like to discuss the required revisions further. Thank you for the opportunity to review your work.

Best regards,

Maria Laura Balestrieri
External Editor
Communications Earth & Environment

Carolina Ortiz Guerrero, PhD
Associate Editor
Communications Earth & Environment

EDITORIAL POLICIES AND FORMAT

If you decide to resubmit your paper, please ensure that your manuscript complies with our editorial policies and complete and upload the checklist below as a Related Manuscript file type with the revised article:

Editorial Policy Policy requirements
(Download the link to your computer as a PDF.)

- Behavioural and social science
- Ecological, evolutionary & environmental sciences
- Life sciences

<https://www.nature.com/documents/nr-reporting-summary.zip>

For your information, you can find some guidance regarding format requirements summarized on the following checklist: (<https://www.nature.com/documents/commsj-phys-style-formatting-checklist-article.pdf>) and formatting guide (<https://www.nature.com/documents/commsj-phys-style-formatting-guide-accept.pdf>).

REVIEWER COMMENTS:

Reviewer #1 (Remarks to the Author):

The manuscript by Zhang et al. presents thermoluminescence (TL) data from four dolomite clast samples and two samples of dolomite bedrock from the exhumed footwall of a normal fault, the Monte Marine fault, in Italy. Dolomite TL is a novel technique, and – as the authors highlight in this contribution – development and application of this analytical approach are important for quantifying the low-temperature cooling, and thus erosion, history of carbonate rocks. In addition, the authors argue that owing to the saturation dose for dolomite, TL of these materials is sensitive to dates up to 10 Ma and slow cooling rates (~5 C/Ma). The authors also suggest that this technique has the potential to bridge to fault slip rates derived from paleoseismology.

I appreciate that the authors are tackling the challenging problem of deriving thermal histories (i.e., cooling histories) from carbonate rocks, advancing the novel technique of TL, and opening new avenues of inquiry with respect to low-temperature thermochronometry, tectonics, structural geology, and earthquake science. In addition, the figures are well-constructed. However, the organization, content, and logic underpinning the present manuscript are collectively lacking. As a reader and reviewer, I am struggling to follow key elements of the paper – from the analytical approach to the results and interpretations. Below I outline my major comments and concerns, and minor comments for the authors to consider.

METHODS: My overarching concern is that it is not possible to understand what the authors did from the main text. This is acutely important in a paper that purports to be “the first” of something. The authors state that TL signal characteristics have been investigated previously and provide citations, and that dolomite has a strong saturation dose, but no additional

concepts or details are provided. In essence, there are no methods in the main text, not even a skeletal mention of the analytical or computational approach to calculating TL dates. Said differently, it is not clear how the authors arrived at the results reported in lines 99-103. The broad readership of Nature Communications Earth and Environment is likely not familiar with TL or OSL, what a 360 C TL peak is, or how the dates (or ages) are calculated. I recognize that this is a short format journal, which presents length-limit challenges and choices, but there is way to construct this paper that judiciously uses space so that some methods are included in the main text, rather than being whole-sale placed at the end of the manuscript in the additional information.

Related, in lines 104-105, the authors state “Cooling history was modeled with different cooling rates until the modeled TL signal intensity (n/N , the relative trap saturation level) was the same as the measured one.” How were these time-temperature histories modeled? What is the relative trap saturation level? How is this measured? In addition, maybe I am missing something, but in lines 123-125 seems like circular logic and should be clarified. Overall, the methods and modeling content that is currently relegated to the supplement is required reading to begin to make inroads on understanding the manuscript and, as such, is a signal that this contribution must be expanded in a longer format journal.

I agree that the concept of closure temperature (T_c) (Dodson, 1973) is complex and appreciate the authors citing Guralnik et al. (2013), but I am unsure why the authors define the closure temperature (T_c) as they do – why 50% of the open system? Why not 90%? How do the results change if you consider a closed system, 90%, as is common in other thermochronology studies. Related, what is “self-examination”? (line 118). Figures 3 and 4 are important for the cooling history and T_c , but the discussion of these figures in the main text is limited and could be strengthened.

SIGNIFICANCE AND NOVELTY: Much of the discussion is centered on the tectonic evolution of the central Apennines region and relationship between earlier thrusting and superimposed extensional tectonics and exhumation. The resulting TL data and inferred cooling rates overlap with regional apatite (U-Th)/He and apatite fission-track thermochronometry (lines 131-133) – so what new insights are gained from this study? I found myself waiting for process-based or more global significance to this work beyond the future applications of dolomite TL for quantifying thermal histories in more slowly cooled carbonate rocks. In lines 169-170, the authors invoke the importance of this work for seismic hazards. This is in part because the Monte Marine fault is an active normal fault, in the vicinity of the 2009 L’Aquila earthquake, and the inferred exhumation rate from the dolomite TL overlaps with the range of rates derived from paleoseismology (lines 162-167). But *how* exactly does the long-term exhumation rate derived from dolomite TL help us understand seismic hazards? This is not clear. Is there some link to deformation during the earthquake cycle? Do the authors envision that slip is happening during coseismic deformation? Or during the interseismic period? I think it is challenging to discern between these with the data in hand and maybe additional data would help “assess the spatio-temporal variations in fault slip rates” (lines 169-170). Nevertheless, considering the broad audience of this journal, I would like to see the authors advance the significance and importance of this work, if possible.

Lines 20, 99-101: What underpins this level of precision in the dates? Related, what underpins the level of precision in the cooling rates? These rates seem overly precise given that the authors are assuming monotonic cooling (lines 304-305), and that likely is an oversimplification.

Line 28: “exhumation/cooling” – these are two different things, but yes are related. I would say “cooling and thus exhumation”

Lines 51-52: Interesting. What are the major element data for the samples reported in this study? How does it impact the TL signal?

Line 53: What is the high saturation dose?

Lines 54-56: This is intriguing, but what do the authors mean by “tectonic/thermal marker”? Does this relate to coseismic temperature rise during fault slip? And impact on the dolomite TL signal?

Line 65: What does “chain” refer to?

Lines 69-70 and elsewhere: How does seismicity link to the longer-term history of extension? Do the authors envision that the exhumation rate is controlled by coseismic deformation or interseismic deformation?

Line 78: What does “hit by” mean? Strong ground motions? What is the spatial relationship between the sample sites and the epicenter?

Lines 82-91: Although this information is interesting for a regional tectonic study, I think that it could be streamlined to make space for methods information.

Lines 90-91: “sub-vertical zones of diffuse extensional shearing” – this is unclear. And what is the relationship between this phenomenon (which implies to me limited slip) and the earthquake cycle?

Lines 91-94: Given that the authors do not return to this concept in the manuscript, I do not think this text is needed.

Line 96: Why are there a header and subheaders here but not elsewhere (i.e., earlier) in the manuscript?

Lines 99-102: Later in the manuscript, specific sample IDs are provided when discussing cooling rates. Sample IDs should

be linked to these dates so that the reader can more readily follow the discussion of the results.

Lines 132-133: The authors state that their data agree with conventional thermochronology data, but in detail, the TL data from the clasts is ~2.5 Ma and this statement suggests there is a pulse of exhumation *after* 2.5 Ma. If the Tc of the dolomite TL overlaps with the apatite (U-Th)/He system, then I am not sure (1) how the targeted rocks are preserved and have not been eroded away by this later exhumation, of (2) if this discrepancy can be resolved by subtle differences in the timing and magnitude of exhumation surround the AIB.

Line 147: In some places this is MMF, Monte Marine Fault, or MMFS. It helps to be consistent.

Reviewer #2 (Remarks to the Author):

Dear Authors, dear Editors,

Please find below my evaluation concerning the manuscript by Zhang and co-authors entitled "Dolomite thermochronometry applied in central Apennines: a novel tool for tectonic reconstruction in carbonate rocks" (manuscript COMMSENV-24-1690-T). First of all, I sincerely apologize for the delay in sending my evaluation.

This manuscript investigates the potential of thermoluminescence (TL) dating on dolomite for application in thermochronometry. The authors provide a new series of data from the central Apennines as test for their novel approach. They propose a detailed analytical protocol to reveal the suitability of the TL dating on dolomite and its application for thermochronometry, with nice analytical results and high saturation doses that are highly valuable for slowly-exhuming regions compared to other OSL/TL techniques. The output TL ages reveal Late Pliocene exhumation history, that is placed in a broader context of the study region.

This is a very interesting manuscript, overall well-written and illustrated. It provided the first attempt to assess the potential of dolomite TL dating in thermochronometry, and as such this is a novel piece of work and important contribution with broad interest and potential of applications in the field of tectonics/geodynamics, especially in carbonate regions where other thermochronometric techniques are lacking. The detailed analytical results set the foundation of this novel technique, which would be further investigated to fully reveal the potential and possible limitations/complications. The application to the central Apennines is also interesting and provides quantification for the late Pliocene to Quaternary extensional tectonics associated to major earthquakes and hazards. Some of the output results and tectonic interpretations are not entirely clear and may be reworked or discussed for clarity. In addition, the significance of the TL ages as exhumation ages or rather related to direct fault activity (fluid circulation, re-crystallization or shear heating) should be better addressed in the discussion.

I have outlined below my questions and suggestions in a set of general and specific comments below.

General comments:

1 – Dolomite TL ages: the authors propose that TL ages on dolomite reflect the rock exhumation and associated cooling during fault activity. The dolomitization processes are exposed in the main text, but this is not entirely clear when this occurred compared to fault activity. In some places along the text it is mentioned that dolomite can be dissolved/recrystallized (joints? Fig. 2b), so I am wondering whether the obtained TL ages could alternatively represent other processes than rock exhumation in that context: (1) dolomite (re-)crystallization, or (2) potential re-heating during fault activity (shear heating or fluid circulation). Such processes have already been observed in other context and for other methods (U-Pb calcite, apatite U-Th/He, OSL/ESR...), so I think that this would be very interesting for readers to further discuss that in the main text and provide more observations/arguments to support the proposed interpretation of dolomite TL ages as constraints for rock exhumation.

2 – TL age difference clasts/bedrocks: there is 0.9Ma age difference between the clasts and bedrock samples (line 103), but this difference is not really well explained in the main text. In Fig. 5 it seems that this is the result of westward propagation of the extension, but given the spatial proximity of the samples (distance/elevation) this seems questionable. Another alternative explanation could (possibly) be sample re-heating for the clasts by fault activity and shear-heating or fluid circulation (linked to my previous comment 1), could this be realistic?

3 – Self-examination approach: I am not entirely sure that I follow and find useful the proposed self-examination approach proposed in the main text for cooling rate calculation (equ. 1, lines 119-126). I totally follow the author's modeling approach and Tc definition (trapping/detrapping balance) which is nicely represented by Fig. 3c. However, deriving an output cooling rate from equ. 1 appears somehow circular if compared to the output cooling rate from modeling (Fig. 4b), given that Tc is an output of the modeling approach (so the two cooling rate estimates are not independent). Maybe I missed something, please consider clarifying this or removing this point.

4 – Analytical results: I appreciate the extent of methodological details and data reported in the Methods section and Supplementary Materials. However, in the main text some of the important information may be missing for clarity or not really referenced, which is making sometimes difficult to follow the results and quality of data. For instance, would it be possible to show some of the TL spectra and MAAD dose-response curves in the main text for clarity (maybe one representative sample)?

Specific comments, by line number:

- Line 1. "Dolomite thermochronometry...". I would suggest to add "TL" or "Thermoluminescence" for clarity in the title.
- Line 19. "dolomite clasts in the fault zone and dolomitized bedrocks". See my general comment about possible (re-)crystallization of dolomite.
- Line 23. "exhumation/cooling history". Are these TL ages reflecting really the cooling history, or rather a (re-)crystallization of re-heating event (fault shear heating)? See my general comment.
- Line 27. Maybe add section titles for clarity, such as "Introduction/Context" there.
- Line 29. "such as fission-tracks and (U-Th)/He methods...". There have been some attempts to use U-Th/He dating on carbonates in the literature (e.g. Cros et al., 2014 GCA...), maybe provide some references there.
- Line 33. "to unravel the thermal/cooling histories of rocks 2-8". I appreciate the amount of references cited there, but I would suggest to cite them progressively along the following lines as they do not provide the same information/success in trapped-charge application for thermochronometry.
- Line 42. "their luminescence and ESR signals". Of what (quartz/feldspar)? Please correct.
- Line 48. "structurally-controlled dolomitization". Yes thus the TL ages on dolomite may also constrain (re-)crystallization and not necessarily cooling histories, especially in fault areas. See my general comment.
- Line 51. "its application in thermochronometry has never been explored". The application of TL in calcite has been explored by Ronca and Zeller (1965), maybe this would be interesting for readers to compare the TL signal characteristics between calcite/dolomite.
- Line 53. "a strikingly high saturation dose". Please provide a range/order of magnitude for readers.
- Line 54. "to be stronger than calcite". To what? Mechanical deformation? Other? Please specify.
- Line 61. "pre-orogenic dolomitized bedrocks". I don't know the study area, but given this information can the authors exclude potential re-crystallization during fault activity and associated fluid circulation? Same comment when reading line 82 "evidence of diffuse dolomitization". Please clarify.
- Line 69. "age of onset of the extensional tectonics". Is there any study using calcite U-Pb dating on fault surfaces for this area? I would also strengthen there that beyond the onset age of fault activity, the aim of TL thermochronometry would be to provide exhumation rates indeed.
- Line 89. "shear deformation". Can fault shear heating may also influence the TL signal, as this has been reported/suggested for other luminescence signals (e.g. Bateman et al., 2012 QG, Lavé et al., 2023 Nature, Tsukamoto et al., 2024 Earth Planets & Space, Heydari et al., 2024 QG...)? I think this is an important point to introduce in the text and to discuss in the light of obtained results...
- Line 92. "dolomite grains (present either as secondary mineral in the carbonate bedrock or clasts in cataclases)". This is unclear whether these investigated grains are inherited from pre-orogenic dolomitization of neofomed during fault activity. Please clarify and provide some arguments for this important point.
- Line 96. There is no "Methods" section in the main text (this comes afterwards at the end), but maybe a quick sentence referring to the methodological section at the beginning of the results section could help the reader to follow.
- Line 101. "at different distances from the main fault traces". How far? Please provide some numbers there (I have the impression that those samples are not that far from the faults, correct?).
- Line 113. "which the electron detrapping rate is half of the trapping rate". Please refer to the methods section where these terms are defined.
- Lines 115-117. I don't really get the usefulness of calculating another cooling rate from the T_c (equation 1), which is only a rough estimate compared to the modeling approach you employed to derive cooling rates presented in the previous paragraph (lines 108-110). Please clarify or consider removing this (see my general comment on this).
- Line 128. "50-70°C". The range can even be larger, see Ault et al. (2019, Tectonics) for discussion and estimates.
- Line 130. "a geothermal gradient of 25°C/km". Are there some estimates for this area?
- Line 133. "showing a major erosion/exhumation event". Please provide some rates there.
- Line 134. "the characteristic saturation dose (D_0) values range 2500–6000 Gy". This is an important result, and could go before when giving the TL ages I think.
- Line 135. "taking a $2D_0$ as maximum". I agree this is a standard calculation, at least in luminescence, but maybe add a reference there for readers (for instance Wintle 2008 Boreas?).
- Lines 137. I would suggest to provide range estimates for maximum TL ages and associated cooling rates.
- Line 143. "dolomite thermochronometry significantly broadens...". How about potential fading, and/or variability in thermal kinetic parameters? I agree some of these are presented in the Methods section, but I would encourage the authors to discuss these further in the main text. This seems important when proposing a new approach.
- Line 147. "at the footwall of the MMF". This fault system is complex, with multiple fault segments so can the exhumation/faulting history be more complex than presented in Figure 5?
- Line 148. "were exhumed to near-surface conditions...". Not correct, the samples were exhumed below the T_c but where still at 2-3km depth at that time, no?
- Lines 155-160. I would disagree with these sentences and interpretation. No the Late Pliocene TL ages reported in this study do not constrain the onset time for exhumation/cooling, they quantify the time at closure depth/temperature but the exhumation could have been ongoing for longer time (samples were at greater depths). Please rephrase and correct this paragraph accordingly.
- Line 162. "crustal extension progressed westward". Is this evidenced by the TL age pattern, or combined with other data? Please clarify, from the TL ages the pattern is reduced (few hundreds of m) and not that clear, no?
- Line 169. "dolomite TL technique". Replace by "dolomite TL dating".
- Line 169. "spatio-temporal variations in fault slip rates". Unclear, the output results rather suggest no variations in fault activity. Maybe rephrase.
- Line 173. "provides the only viable and effective approach...". This sounds a bit strong to me and seems not necessary. The new dolomite TL ages presented there are interesting and highlight the potential of the method for thermochronometry, but I would suggest to rephrase and propose the potential (and stress the wide application) of this new method, to be confirmed

by further analysis and case studies.

- Line 174. "cooling/exhumation history". Also for fault activity, no?
- Line 178. "four dolomite clasts". What is the size of samples?
- Line 180. "far away from the fault". Please provide numbers there for distance to the fault.
- Line 181. Why only clasts and no bedrock were analyzed by XRD? Can this be possible that bedrock samples differ in mineralogy?
- Line 189. "5mm thickness". Is this enough to ensure no light exposure and TL signal removal for these outcrop samples? Usually for feldspar/quartz trapped-charge dating applied to thermochronometry, 1cm of the outer surfaces are removed (e.g. King et al., 2016 ChemGeol). Could this have an influence on the TL results, given the potential light penetration in clear dolomite/calcite crystals?
- Line 191. The analysed grainsize is reported after sample preparation, but what is the initial grainsize for dolomite crystals? Have the authors made and looked at thin section to evaluate the dolomite grainsize? This can be much smaller, and implication for dose rate calculation?
- Line 232. "peak shifting method". Please provide reference(s) there. Same comment for line 238 (E and s values can be deduced...).
- Line 240. "Table S3". I think that the estimated lifetimes should be discussed, first to state the relative stability of the TL360°C peak, and also I would encourage the authors to discuss the potential usefulness of the TL285°C (lower thermal stability, so complementary to the other peak for tighter constraints on the late-stage cooling history), although practically at present this peak cannot be used (but maybe future work will allow to use it?). Please consider adding few lines about this.
- Line 252. I am not completely convinced by the fading results presented in figure S7. First there seems to be an apparent peak shift measured in the 360°C region, is it just a visual impression or real? And in addition, would it be possible to present the fading results centered on the region of interest, i.e. the 340-380°C region, to better evaluate the results? In any case, I would encourage the authors to add few words in the main text about this, as well as for the thermal kinetic parameters.
- Line 255. Where are the U-Th-K contents reported? Please provide the data in text or as table.
- Line 290. "the samples were far below the surface". How far? In the sample collection section, these are reported to be sampled on the fault plane, not at depth. Please clarify, given that cosmic rays can penetrate as deep as meters this may not be negligible to consider this effect on sample dose rates.
- Line 293. "assuming first-order kinetics". Is this a valid assumption? From Fig. S6 I would say yes, but this should be discussed and possibly are there any references supporting this point?
- Line 295. How is n/N defined from the measurements? In table 1 this is defined as the "natural TL signal intensity", but this is not clear. It should be linked somehow to the parameters of the dose response curve fitting (Fig. S3) but no indication are given, please clarify.
- Line 295. This is not clear that equation 1 is in fact 2 equations next to each other, maybe consider showing them on two lines for clarity.
- Table 1. Please provide a clearer definition of n/N , and refer to text and equations for T_c ("with our definition" is not clear).
- Figure 2. In panel a, what are the letters a,b,c on the block diagram? In the stereoplots, the colors are not clear on my version, maybe try to update them for clarity.
On panel d, is it possible to have the sampled surface indicated on the picture?
- Figure 3. On panel c, what are the different temperature-lines shown? Every 10%? Please describe this in caption, and maybe indicate the 10-50-90% lines in bold.
In caption, the panel a and b differ (cooling history modeling vs. cooling rate modeling), please correct for consistency.
- Figure 4b. See my general comment, I am not fully convinced by the self examination procedure and results reported in Fig. 4b.
- Figure 5. Some comments/suggestions for this figure: The isotherm showed on the figure is not correct, it should first follow the topography and is perturbed by the fault advection. Please correct.
In panel a, no exhumation is considered but the data cannot constrain this (since rocks are above the T_c), so there might be exhumation/fault activity already before no?
In panel b, what is the timing and importance of the thrust cataclasite? This is not discussed in the main text...
Finally, in panel c is illustrated the fault propagation to explain the ~0.9Ma difference in TL ages. Is it the case (I could not figure out this explicitly in the text, or I missed it) and is it realistic given the proximity in samples (both in distance and elevation)?

I hope these comments and suggestions may be useful for revising this interesting manuscript.

Sincerely,
Pierre Valla
(Grenoble, 26 July 2024)

Reviewer #3 (Remarks to the Author):

Comments to the manuscript:

Title: Dolomite thermochronometry applied in central Apennines: a novel tool for tectonic reconstruction in carbonate rocks.

Authors: Zhang et al.

The authors present the results of a new thermochronological tool to assess the exhumation/cooling history of carbonate rocks based on the dolomite thermoluminescence (TL) dating.

As claimed by the authors, the results of these studies could have a very important impact on the reconstructions of the exhumation/cooling history in carbonate rock regions which until now have suffered the lack of appropriate thermochronological methods due to the absence of target minerals (e.g., zircon, apatite, quartz, feldspar). Furthermore, since the TL signal of dolomite has a strikingly high saturation dose, the method could bring a great advantage in thermal history reconstruction in regions with low cooling rates (i.e., as low as 5 °C/Ma), also providing a useful tectonic/thermal marker to investigate long-term crustal deformation in the brittle seismogenic crust.

Therefore, the manuscript represents a novel and valid contribution to the Earth Sciences community and is eligible for Communications Earth & Environment journal.

The various sections presented in the "Methods" chapter are written in a very thorough and detailed way. However, I believe that a very concise sentence at the end of each section could help even readers who are not specialized in the techniques used to understand the fundamental aspects of the method, the various critical issues encountered and how they were overcome.

Regarding the case of application of the novel thermochronometry method in the Central Apennines, three main points of criticism arise, which I believe should be considered to make the work suitable for publication in "Communications Earth & Environment".

1) My main criticism is aimed at the fact that the exhumation values evaluated for the Monte Marine area from the Late Pliocene to Present day are always discussed with regard to the MMF activity alone, so much so that throughout the manuscript the presence and contribution to the exhumation of rocks due to the "regional uplift" of the Apennines is never discussed.

Considering for example the very recent paper by Racano et al. (2024), where there are also common co-authors with the present manuscript, it is argued that: ... "Thermochronological and geochemical studies provide evidence for the recent rapid uplift in the Central Apennines. Apatite fission track and (U-Th)/He cooling ages in the Central Apennines reveal that the most recent exhumation phase began at approximately 2.5 Ma, coinciding with the onset of normal faulting and the opening of intra-montane basins throughout the Central Apennines (Cosentino et al., 2017; Fellin et al., 2021)".

I believe, therefore, that it is necessary to indicate in the Introduction section also the presence of the contemporary crustal process of the uplift of the Central Apennines, and perhaps discuss in the next part of the manuscript the possibility of discriminating and evaluating the extent of exhumation associated with the local activity of the MMF compared to the regional uplift of the Central Apennines.

Still regarding the aforementioned issue, I also ask whether the hypothesis of analyzing a dolomite sample (if present) also in the hanging wall block of the MMF has been evaluated, considering that the bedrock is in outcrop both towards the most northwestern and southeastern portions of the fault (e.g., CARG Sheet 348 "Antrodoco", ISPRA, at https://www.isprambiente.gov.it/Media/carg/348_ANTRODOCO/Foglio.html).

2) The authors argue that the exhumation rates obtained through the dolomite TL dating (up to 0.8–0.9 mm/yr, see line 163-167) are compatible with the paleoseismic estimates of fault slip rates in the central Apennines, which are in the order of 0.4–1.2 mm/yr (preferred 0.6–0.8 mm/yr). However, I suggest that it might be more correct to compare the exhumation rate obtained by dolomite thermochronometry with the "downthrow-rate" associated with the MMF, especially considering that the main fault plane often exhibits a "quite low-angle" geometry (i.e., 40-45°, in the Barete area) as it seems to be observed from the stereoplot of fig. 2a.

3) In the schematic 2D evolutionary model of Fig. 5 it is shown that from about 3.4 to about 2.5 Ma (so at least for about 0.9 Ma) only the easternmost fault plane of the MMF is active - that is, the one between the TL SAMPLE (bedrock) and the TL SAMPLE (Cataclasite), then faulting progressed westward post c. 2.5 Ma. In this hypothesis, considering a "slip-rate" of about 0.6-0.8 mm/yr (see line 167) this segment should have accumulated a slip of 500-700 m. Do the authors have geological evidence in the field of this easternmost plane with such values of cumulative slip?

You should feel free to use my comments as they feel appropriate. I hope they are useful. Thank you for considering me as a reviewer for this manuscript.

Very Truly Yours

Communications Earth & Environment is committed to improving transparency in authorship. As part of our efforts in this

direction, we are now requesting that all authors identified as 'corresponding author' create and link their Open Researcher and Contributor Identifier (ORCID) with their account on the Manuscript Tracking System prior to acceptance. ORCID helps the scientific community achieve unambiguous attribution of all scholarly contributions. You can create and link your ORCID from the home page of the Manuscript Tracking System by clicking on 'Modify my Springer Nature account' and following the instructions in the link below. Please also inform all co-authors that they can add their ORCID to their accounts and that they must do so prior to acceptance.

Version 1:

Decision Letter:

Dear Dr Zhang,

Your revised manuscript titled "Dolomite luminescence thermochronometry: a novel tool for tectonic reconstruction in carbonate rocks with application to central Apennines" has now been seen by our original reviewers #2 and #3, whose comments appear below. In light of their advice we are delighted to say that we are happy, in principle, to publish a suitably revised version in Communications Earth & Environment.

We therefore invite you to revise your paper one last time to address the remaining concerns of our reviewers. Specifically, please comply with the following editorial requests: 1) Expand your discussion to enhance the clarity of the proposed hypothesis, toning down the "link between the onset of D2 and the 2.5Ma apparent age", as well as 2) Reorganize the supplementary material references. At the same time we ask that you edit your manuscript to comply with our format requirements and to maximise the accessibility and therefore the impact of your work.

EDITORIAL REQUESTS:

****Please take care to match our formatting and policy requirements. We will check revised manuscript and return manuscripts that do not comply. Such requests will lead to delays. ****

SUBMISSION INFORMATION:

OPEN ACCESS:

Communications Earth & Environment is a fully open access journal. Articles are made freely accessible on publication. For further information about article processing charges, open access funding, and advice and support from Nature Research, please visit <https://www.nature.com/commsenv/open-access>

Link Redacted

Best regards,

Carolina Ortiz Guerrero, Ph.D.
Associate Editor
Communications Earth & Environment

Maria Laura Balestrieri
External Editor
Communications Earth & Environment

REVIEWERS' COMMENTS:

Reviewer #2 (Remarks to the Author):

Dear Authors, dear Editors,

Please find below my evaluation concerning the revised manuscript by Zhang and co-authors entitled "Dolomite luminescence thermochronometry applied in central Apennines: a novel tool for tectonic reconstruction in carbonate rocks".

I thank the authors for the thorough revisions on the manuscript, which have answered most of my initial comments. I think that the revised manuscript is clearer, especially regarding the proposed methodology and TL approach for thermochronometry and fault activity, which is the central and novel piece of the present work. I still have few comments regarding the data/modeling presentation and discussion of the output results for tectonic reconstruction. My main concern is about the use/referencing of "Methods" section and of the Supplementary Material information in the main text, which is lacking sometimes and prevent the readers to easily follow the methodology or the results. Concerning the tectonic reconstruction, I appreciate the discussion of TL outputs for clasts and the possible interpretation as reflecting faulting events. However, I still think the proposed interpretation (one single reset event) is a strong hypothesis and would require more discussion.

I have outlined below my questions and suggestions in a set of specific comments.

Specific comments, by line number:

- Line 17. "the cooling and exhumation history". Maybe replace history by histories.
- Line 20. "novel thermochronometric technique".
- Line 21. "The study area is located...". Maybe rephrase this sentence beginning to stage that you apply the novel approach to a specific area...
- Line 48. "The growth of the luminescence signal...". Maybe a reference is missing there.
- Line 56. "luminescence dating". "thermochronometry" would better apply there. And same comment for line 58.
- Line 62. "In regions with low exhumation rates". What range is considered there? And above in the text are reported cooling rates, not exhumation rates. Please correct.
- Line 65. "to rock bodies". Maybe "lithologies" or "bedrocks" would be better there, no?
- Line 73. "This method is termed paleothermometry nowadays". Is this sentence useful for the message?
- Line 74. "a strong TL peak above 300 °C,...". Please provide a reference there.
- Line 74. "at the surface temperature". Would this be valid in hot settings, i.e. deserts?
- Line 94. "in the transition". Maybe "during the transition" would read better there.
- Line 95. "D1 orogenic". The deformation phases D1 and D2 are only presented after in the next section (line 119 and Figure 2), so I would suggest to rephrase and be more explicit there.
- Line 133. "Four dolomite clasts". Maybe refer to the Supplementary Material and/or Methods section for sample location (Table S1, etc).
- Line 142. "In case of dating,...". I am not sure the term "dating" is clearly applicable there (same remark for line 143 and later in the text), since the purpose is not really dating but thermochronometry...maybe check and rephrase.
- Line 144. "A multiple-aliquot additive-dose dating protocol". A reference is missing there.
- Line 149. "TL ages". I would recommend to add "apparent" before age since this is a closure age (as discussed after).
- Line 154. "For trapped charge dating". I would suggest to replace dating by thermochronometry.
- Line 183. "E varies in the range of 1.71–1.88 eV, logs in the range of 13.0–14.6, and D0 in the range of 2450–5920 Gy.". Please refer to Supp. Material and figures for these results.
- Line 185. "a medium E of 1.80 eV". What is a "medium" value? Average? Median? Please clarify.
- Line 189. As for my initial review, I still don't see the point of calculating two Tc values, since they are similar (Tc is extracted from the cooling rate, and corresponding roughly to Tc_half as written on lines 177). So there is no real meaning to compare the two (Fig. 4b), or I have missed something...please clarify or remove this sentence and comparison.
- Line 201. "This evidence suggests a closed chemical system". Yes I agree the patterns are similar between bedrocks and clasts in Fig. S6, but the clasts are clearly enriched in all REY so would it be plausible to have an homogenous enrichment? Please clarify.
- Lines 207-210. "TL ages of the fault clasts are nearly coincident, regardless of their sizes and the along-strike locations within the fault damage zone. Thus, the ~2.5 Ma TL ages of the dolomite fault clasts likely indicate a complete resetting event induced by shear heating during development of the MMF".

I would somehow disagree with this strong statement about the significance of the 2.5Ma TL age for clasts. First, if the block exhumation is similar to the bedrock samples, we need to consider that at ~2.5 Ma the clasts were also still at depth. With the bedrock cooling rate of ~11°C/Ma, this relates to ~37.5°C so the thermal detrapping rate may not be zero. From Figure 2f this can be small, but this first needs to be evaluated/discussed by authors.

Second, I agree that the overall very similar age around 2.5 Ma for all clasts is a nice results and should be better emphasized as reflecting a similar trapped-charge history for these samples, but this could also be several small resetting events that have accumulated over the exhumation history of the clasts...this needs to be better discussed I think.

- Line 232. "an average exhumation rate of 0.45 mm/yr". Please propagate the uncertainties on cooling rates for the exhumation rates. This will allow the readers to better compare with literature estimates.

- Line 237. "average: ~2.5 Ma". What is the uncertainty that can be derived from all the clasts? This could then be propagated to the slip rate estimate, no?

- Line 238. "indicate a fault-related signal resetting event". See my comment above for lines 207-210. This apparent age can also be the sum of small multiple events, so the link between onset of D2 and this apparent age is a strong hypothesis. I would recommend the authors to tune down this and/or explain why they think this hypothesis could be valid (independent observations or data?).

- Line 814. Figure 3. There is a lot of information on this figure, panels are small and we cannot read text in panel c. I would recommend the authors to split this into two figures, one for modeling (d-e-f) and one for TL analysis (a-b-c).

- Line 834. Figure 4. As stated above, I don't think that figure 4b is insightful, compared to the figure 4a is important and valuable.

- Supplementary material: This document compiles a lot of technical information that are important for the study and TL thermochronometric approach, but there is no real structure and only figures and tables are provided, which may not help the readers. Maybe re-structure the document with different sections, starting with sample information, TL analysis, modeling etc. At present this is difficult to follow it for (non-expert) readers...

I hope these comments and suggestions may be useful for finalizing the revisions for this manuscript, and that they could make the authors message clearer.

Sincerely,
Pierre Valla
(Grenoble, 5 February 2025)

Reviewer #3 (Remarks to the Author):

Dear Editor and dear Authors,

I read with pleasure the revised manuscript titled "Dolomite luminescence thermochronometry: a novel tool for tectonic reconstruction in carbonate rocks with application to central Apennines" by Dr Zhang and colleagues.

I really appreciated the Authors' effort in considering most of the comments made by the reviewers. I believe that this has brought a concrete benefit not only in the presentation of the innovative geochronological method, but also in its application example to the tectonic context of the Apennines.

With respect to this last point, the new evolutionary model now proposed that, thanks also to the new TL dolomite ages, indicates a coseismic signal resetting event at c. 2.5 Ma, most likely due to the frictional heating at the onset of the D2 post-orogenic extension (rather than cooling/exhumation ages), provides a very stimulating hypothesis that also addressed my first question about the problem of discerning the exhumation associated with the local activity of the MMF compared to the regional uplift of the Central Apennines.

Congratulations, also, about the new figure 5 that schematizes in a very clear and effective way the results of the study.

Finally, concerning the compatibility of the MMF slip rate values obtained in the present work with the short-term ones (lines 246-248), I suggest the authors also consider the recent paleoseismic study by lezzi et al. (2023) who estimated an average slip rate value of about 1.0 mm/yr for the last 35 kyr of MMF fault activity.

lezzi, F., et al. (2023) - Slip localization on multiple fault splays accommodating distributed deformation across normal fault complexities. TECTONOPHYSICS. - ISSN 0040-1951. - 868:, p. 230075. [10.1016/j.tecto.2023.230075]

Therefore, I hereby inform you that I am fully satisfied with the Authors' response and the revised version of the manuscript. I hope that my contribution has been useful to the Authors and thank you again for considering me as a reviewer for this manuscript.

Best Regards

To: CEE Editors (Dr. Maria Laura Balestrieri and Dr. Carolina Ortiz Guerrero),

Re: Response to reviewers' comments on **COMMSENV-24-1690-T**, entitled '*Dolomite thermochronometry applied in central Apennines: a novel tool for tectonic reconstruction in carbonate rocks*', by Junjie Zhang, Giorgio Arriga, Federico Rossetti, Valentina Argante, Dennis Kraemer, Domenico Cosentino, Paola Cipollari, Sumiko Tsukamoto

Dear editors and reviewers,

We would like to express our sincere gratitude for your efforts in providing insightful and constructive feedback to the manuscript. We warmly appreciate all the comments from all reviewers that were of great help in improving the scientific message of the manuscript and clarify key aspects in our reconstruction.

We have revised our manuscript following the reviewers' comments. Below, we provide a detailed point-by-point response to your comments.

To make it convenient for reading, we make our reply in blue.

REVIEWER COMMENTS:

Reviewer #1 (Remarks to the Author):

The manuscript by Zhang et al. presents thermoluminescence (TL) data from four dolomite clast samples and two samples of dolomite bedrock from the exhumed footwall of a normal fault, the Monte Marine fault, in Italy. Dolomite TL is a novel technique, and – as the authors highlight in this contribution – development and application of this analytical approach are important for quantifying the low-temperature cooling, and thus erosion, history of carbonate rocks. In addition, the authors argue that owing to the saturation dose for dolomite, TL of these materials is sensitive to dates up to 10 Ma and slow cooling rates (~5 C/Ma). The authors also suggest that this technique has the potential to bridge to fault slip rates derived from paleoseismology.

I appreciate that the authors are tackling the challenging problem of deriving thermal histories (i.e., cooling histories) from carbonate rocks, advancing the novel technique of TL, and opening new avenues of inquiry with respect to low-temperature thermochronometry, tectonics, structural geology, and earthquake science. In addition, the figures are well-constructed. However, the organization, content, and logic underpinning the present manuscript are collectively lacking. As a reader and reviewer, I am struggling to follow key elements of the paper – from the analytical approach to the results and interpretations. Below I outline my major comments and concerns, and minor comments for the authors to consider.

METHODS: My overarching concern is that it is not possible to understand what the authors did from the main text. This is acutely important in a paper that purports to be “the first” of something. The authors state that TL signal characteristics have been investigated previously and provide citations, and that dolomite has a strong saturation dose, but no additional concepts or details are provided. In essence, there are no methods in the main text, not even a skeletal mention of the analytical or computational approach to calculating TL dates. Said differently, it is not clear how the authors arrived at the results reported in lines 99-103. The broad readership of Nature Communications Earth and Environment is likely not familiar with TL or OSL, what a 360 C TL peak is, or how the dates (or ages) are calculated. I recognize that this is a short format journal, which presents length-limit challenges and choices, but there is way to construct this paper that judiciously uses space so that some methods are included in the main text, rather than being whole-sale placed at the end of the manuscript in the additional information.

Thanks a lot for the comment! We acknowledge that the majority of readers of the CEE journal may not have the background knowledge of luminescence dating. In the revised manuscript, we have moved the necessary information from the 'Methods' to the main text to explain how the luminescence age is obtained and how the cooling rate is modeled. Please refer to lines 41-53, 151-163 of the revised typescript for the details.

Related, in lines 104-105, the authors state "Cooling history was modeled with different cooling rates until the modeled TL signal intensity (n/N , the relative trap saturation level) was the same as the measured one." How were these time-temperature histories modeled? What is the relative trap saturation level? How is this measured? In addition, maybe I am missing something, but lines 123-125 seems like circular logic and should be clarified. Overall, the methods and modeling content that is currently relegated to the supplement is required reading to begin to make inroads on understanding the manuscript and, as such, is a signal that this contribution must be expanded in a longer format journal.

In the revised manuscript, we have added all the necessary information for the cooling rate modeling in the section of 'Dolomite TL ages and cooling rates'.

The relative trap saturation level (n/N) equals to the luminescence signal saturation level (I_n/I_0). For a collected sample, we can measure its natural TL intensity, the I_n . Meanwhile, we artificially irradiate the sample (prepared with aliquots) with different doses by a beta source (^{90}Sr) in the lab and measure the corresponding TL intensities. From the relationship of TL intensity and dose, we can fit the growth curve with a function [$y=I_0*(1-\exp(-(x+D_e)/D_0))$]. From this function, we got the maximum TL intensity at the saturation level, the I_0 . The signal saturation level is I_n/I_0 .

With a certain cooling rate, we can model the signal growth with time, from an initial temperature of 100 °C (a temperature so high that no electrons can be stored in the traps) to the surface temperature at L'Aquila (10 °C). With a different cooling rate, the signal saturation level of a sample exhumed to the surface will be different. Since we measured the I_n/I_0 of the surface sample, we can deduce its cooling rate.

From the growth curve function, we can also get the D_e , which is the dose the sample received in nature. By measuring the radioactive elements (U, Th, K) concentrations, we can calculate the annual dose (\dot{D}). The TL age can be calculated by D_e/\dot{D} . We have added these information in the introduction part.

Regarding the lines 123-125 in the original manuscript about cooling rate self-examination. We have replaced this part, with more detailed discussion on T_c . The detailed changes are listed under the next question regarding the T_c (see lines 169-191 of the revised typescript).

I agree that the concept of closure temperature (T_c) (Dodson, 1973) is complex and appreciate the authors citing Guralnik et al. (2013), but I am unsure why the authors define the closure temperature (T_c) as they do – why 50% of the open system? Why not 90%? How do the results change if you consider a closed system, 90%, as is common in other thermochronology studies. Related, what is "self-examination"? (line 118). Figures 3 and 4 are important for the cooling history and T_c , but the discussion of these figures in the main text is limited and could be strengthened.

We have revised this part substantially to make it read more logically (see lines 169-191 of the revised typescript).

For the TL signal of dolomite in this study, the 50% of the open system is more suitable for defining a closure temperature than 90%. The reason is as below:

Since we know the cooling rate, the apparent TL age and the surface temperature (T_s), we can directly calculate the closure temperature (T_{c_cal}) mathematically from the equation: cooling rate = $(T_c - T_s)/\text{age}$. This T_{c_cal} is the true T_c as it strictly follows the mathematic definition of T_c .

The T_{c_cal} values of the two bedrock samples (LUM4774, LUM4775) are 64.0 and 59.1 °C, respectively. With a certain cooling rate, we know the change of detrapping/trapping rate ratio with temperature. We found that these T_{c_cal} values correspond to a detrapping/trapping ratio of 48 % for sample LUM4774 and 50 % for sample LUM4775. Thus, taking the 50 % of open system seems to be the most suitable approximation.

In the new figure 4, we compared the difference between T_{c_cal} and T_{c_half} under different cooling rates and dose rates. In most cases, their difference is smaller than 0.4 °C.

SIGNIFICANCE AND NOVELTY: Much of the discussion is centered on the tectonic evolution of the central Apennines region and relationship between earlier thrusting and superimposed extensional tectonics and exhumation. The resulting TL data and inferred cooling rates overlap with regional apatite (U-Th)/He and apatite fission-track thermochronometry (lines 131-133) – so what new insights are gained from this study?

We welcome this comment, since it provides the opportunity to emphasise that our first aim is to present the novel dolomite TL thermochronometry and its feasibility for geological/tectonic reconstruction using the Central Apennines as a test site. Actually, the low-T apatite/zircon (U-Th)/He and apatite fission-track thermochronometry methods (conventional thermochronology) are typically limited to siliciclastic bedrocks. One key advantage of using thermoluminescence (TL) dating in dolomite is its applicability in regions where siliciclastic rocks are scarce or absent. This manuscript emphasizes the potential of dolomite TL dating in regions like the central Apennines, which are predominantly composed of carbonate bedrocks. In the central Apennines, siliciclastic rocks are mainly restricted to syn-orogenic foredeep deposits and Pliocene-Quaternary sediments. Such limited occurrence of the siliciclastic rocks in the region restricts the applicability of the conventional thermochronology. Indeed, dolomite TL dating allows to increase the range of thermochronological studies, enabling more detailed reconstructions, in areas where siliciclastic rocks aren't present. We referenced previous studies using apatite thermochronology methods in the central Apennines to highlight that the TL data aligns well with results from conventional thermochronology, confirming its reliability. Following the reviewer's comment, we have modified the Introduction section to better present the aim of our study and the Discussion section has been reorganized to enhance significance of our results for what concerns carbonate thermochronology and its implications when applied to the central Apennines.

I found myself waiting for process-based or more global significance to this work beyond the future applications of dolomite TL for quantifying thermal histories in more slowly cooled carbonate rocks. In lines 169-170, the authors invoke the importance of this work for seismic hazards. This is in part because the Monte Marine fault is an active normal fault, in the vicinity of the 2009 L'Aquila earthquake, and the inferred exhumation rate from the dolomite TL overlaps with the range of rates derived from paleoseismology (lines 162-167). But *how* exactly does the long-term exhumation rate derived from dolomite TL help us understand seismic hazards? This is not clear.

Thank you for this comment. When applying the dolomite technique to the central Apennines, our main aim was to reconstruct the long-term evolution of seismogenic faulting in the region. In fact, a better understanding of both short- and long-term evolution of active fault system can offer valuable insights into the spatio-temporal scales involved in fault nucleation, development and propagation. In this regard, the comparison of the long-term slip rates with those available in the literature for the coseismic fault slip goes should guide to a more complete understanding of how the seismogenic crust has been responding to the tectonic stress field(s) through time (for instance, recognizing seismic transients in the seismic cycle or overall variation from the mean expected values). Tectonic activity reflects the brittle response to both regional and local forces, which are shaped by the tectonic evolution of an area and directly influence seismicity. Understanding this process over both

short- and long-term timescales can offer valuable insights into its evolution from its onset to the present day. In the context of a propagating rift, comparing the current stress regime with earlier stages can reveal whether the rift is likely to progress toward more mature stages, which may involve processes that affect seismicity, such as fault coalescence and linkage.

Is there some link to deformation during the earthquake cycle? Do the authors envision that slip is happening during coseismic deformation? Or during the interseismic period? I think it is challenging to discern between these with the data in hand and maybe additional data would help “assess the spatio-temporal variations in fault slip rates” (lines 169-170). Nevertheless, considering the broad audience of this journal, I would like to see the authors advance the significance and importance of this work, if possible.

Thank you for this comment. The new TL dolomite ages, showing a rejuvenation of the TL signal when moving from the footwall rocks to the fault damage zone (from ca. 4.6 to ca. 2.5 Ma), indicate a coseismic signal resetting event, most likely due to the frictional heating during fault activity, which has also been reported for quartz and feldspar ESR and luminescence signals in previous studies (see lines 193-220 of the revised typescript)

Lines 20, 99-101: What underpins this level of precision in the dates? Related, what underpins the level of precision in the cooling rates? These rates seem overly precise given that the authors are assuming monotonic cooling (lines 304-305), and that likely is an oversimplification.

The precision of a TL age depend on the uncertainty of the D_e measurements, and the dose rate error. Typically, the relative error of a TL age is 5-10 %. In our study, the relative errors of TL ages are in the range of 7-12 %.

The cooling rate error depends not only on the D_e error and dose rate error, but also on the errors of thermal kinetic parameters (E , s). As a result, the relative error of the cooling rate is a bit higher than the TL age. In our study, the relative error of cooling rate is 15-19 %, which is also reasonable.

It is true that in this study we assume the simple situation, a constant cooling rate, for modeling. If the cooling rate changes with time, what we get here will be a mean cooling rate. In the revised manuscript, we have discussed the potential of applying the 295 °C TL peak together with the 350 °C TL peak to study the cooling rates at different temperature regimes. However, so far the essential parameters of the 295 °C TL peak needed for cooling history modeling cannot be measured accurately as this peak is overlapping with a 250 °C TL peak.

Line 28: “exhumation/cooling” – these are two different things, but yes are related. I would say “cooling and thus exhumation”

Revised as suggested. Thanks.

Lines 51-52: Interesting. What are the major element data for the samples reported in this study? How does it impact the TL signal?

In Lines 52-52 of the original manuscript, we stated that ‘Dolomite exhibits a thermoluminescence (TL) signal that is mainly related to the substitution of Ca and Mg by Mn.’ We think that your question is about the relationship between TL single efficiency and Mn concentrations of our samples. In the paper of Medlin (1961), they studied the TL signal of synthetic dolomite with impurities of various ions such as Mn^{2+} , Pb^{2+} , Zn^{2+} , Fe^{2+} , N^{2+} , etc. The results show that the dolomite TL signal is only sensitive to Mn^{2+} . The efficiency of TL (TL intensity with a unit dose of irradiation) initially increases with increasing Mn^{2+} concentrations, and then decreases once the Mn^{2+} concentrations is over 0.002 mole fraction (which corresponds to 600 ppm).

We have summarized the TL efficiency and Mn^{2+} concentrations of the six dolomite samples in our study. The concentrations of Mn are less than 100 ppm for our samples. Generally, a positive

relationship between the TL efficiency and Mn concentration can be observed for both the clast and bedrock, which is consistent with the observation of Medlin (1961).

It is not clear yet why the bedrock samples have an overall higher TL efficiency than the clast samples. However, we want to point out that our XRD analyses show that all the bedrock and clast samples are dolomite, and the TL emission spectra of the clast and bedrock are also similar to each other (Fig S1 and S2 in the revised supplementary file).

Line 53: What is the high saturation dose?

We have added the related information in the introduction part, in lines 47-49.

The growth of the luminescence signal with irradiation follows a single saturating exponential function, as

$$I_n = I_0 * (1 - e^{-\frac{D_n}{D_0}}) \quad (1),$$

D_0 is the characteristic saturation dose. It is a parameter dependent on the specific luminescence signal.

Lines 54-56: This is intriguing, but what do the authors mean by “tectonic/thermal marker”? Does this relate to coseismic temperature rise during fault slip? And impact on the dolomite TL signal?

Rheological tests have documented that dolomite is harder than calcite. Therefore, dolomite may retain the polyphase tectonic history of a geological region. Indeed, the frictional heating during fault slip may reset the TL signal of dolomite. In the revised manuscript we have added much more discussion and attributed the TL ages of dolomite clasts in the fault damage zone to a resetting event during fault movements (see lines 209-238 of the revised manuscript).

Line 65: What does “chain” refer to?

We added “orogenic”. Thanks!

Lines 69-70 and elsewhere: How does seismicity link to the longer-term history of extension? Do the authors envision that the exhumation rate is controlled by coseismic deformation or interseismic deformation?

Seismicity in the central Apennines is linked to the extensional fault pattern, which is described in the previous lines. The intent of this phrase is to highlight the fact that despite the short-term seismic

behavior of these active faults is well known and studied, less is known about the long-term temporal constraints of these structures, particularly when the faults started developing. Indeed, conventional thermochronology is not aimed directly at determining if the deformation occurred during the coseismic or interseismic phase. Although defining the seismic behavior of the structure is a very interesting topic, it is beyond the scope of the manuscript that is instead aimed at obtaining temporal constraints on the cooling/exhumation history of the faulted bedrock (see also the response to the first comment of Rev#3).

Line 78: What does “hit by” mean? Strong ground motions? What is the spatial relationship between the sample sites and the epicenter?

We meant that the epicenter of the 2009 earthquake occurred within the AIB and the area was affected by strong ground motions. The phrase has been changed to include references to earthquakes that directly involved the activation of the Monte Marine fault.

Lines 82-91: Although this information is interesting for a regional tectonic study, I think that it could be streamlined to make space for methods information.

Results presented in this section and shown in Fig. 2 is relevant to support the evidence that the dolomite grains were affected by both D1 and D2 tectonic events, and, therefore, represent a suitable geological marker to record the regional evolution from pre-orogenic times (at least). This scenario is supported by the structural data that we collected in the field. Therefore we believe that it is appropriate describing in detail the tectonic structures. However, following the reviewer’s advice, the necessary information of methods was implemented in the revised text (see lines 41-53, 151-163 of the revised typescript).

Lines 90-91: “sub-vertical zones of diffuse extensional shearing” – this is unclear. And what is the relationship between this phenomenon (which implies to me limited slip) and the earthquake cycle?

The extensional fault zone is characterized by a main fault plane which separates the faulted carbonate bedrock from the Pliocene-Quaternary deposits of the AIB. The fault zone is associated with multiple synthetic and antithetic fault planes, which therefore form a delocalized array of fault planes. It is worth noting that (i) the MMF developed onto a pre-deformed bedrock, which can influence the localization of fault planes, and (ii) the MMF is part of a fault system capable of generating > 6.5 Mw earthquakes, which implies that the rupture zone is wide and formed by multiple faults.

Lines 91-94: Given that the authors do not return to this concept in the manuscript, I do not think this text is needed.

Thank you for this observation. Indeed, following the comment on Lines 82-91, this result is extremely important, as we need to prove that the dolomite provides a geological marker which formed before both the syn- and post-orogenic tectonic phases. Since this manuscript aims to provide a new thermochronological tool to investigate the cooling/exhumation of carbonate bedrocks, it must be demonstrated that the dolomite that accumulated the TL signal crystallized before the tectonic phase we want to investigate.

Line 96: Why are there a header and subheaders here but not elsewhere (i.e., earlier) in the manuscript?

Following the layout format of CEE papers, we did not provide a header at the beginning of the main text, in the original manuscript. To make the review more convenient, we now have added a header of ‘Introduction’ at the beginning.

Lines 99-102: Later in the manuscript, specific sample IDs are provided when discussing cooling rates. Sample IDs should be linked to these dates so that the reader can more readily follow the discussion of the results.

Revised as suggested. Thanks!

Lines 132-133: The authors state that their data agree with conventional thermochronology data, but in detail, the TL data from the clasts is ~2.5 Ma and this statement suggests there is a pulse of exhumation *after* 2.5 Ma. If the Tc of the dolomite TL overlaps with the apatite (U-Th)/He system, then I am not sure (1) how the targeted rocks are preserved and have not been eroded away by this later exhumation, of (2) if this discrepancy can be resolved by subtle differences in the timing and magnitude of exhumation surround the AIB.

We refer to the regional exhumation of the central Apennines, occurring since the Late Pliocene. The erosion varies spatially depending on the different lithologies and mechanisms. The erosion rate in the L'Aquila Basin is very low, and, consequently, the post-Pliocene rock exhumation was mainly related to the tectonic activity of the extensional fault system. This reconstruction is now clearly supported by the documented rejuvenation of the TL dolomite ages within the damage zone of the MMF

Line 147: In some places this is MMF, Monte Marine Fault, or MMFS. It helps to be consistent.

Fixed. Thanks!

Reviewer #2 (Remarks to the Author):

Dear Authors, dear Editors,

Please find below my evaluation concerning the manuscript by Zhang and co-authors entitled "Dolomite thermochronometry applied in central Apennines: a novel tool for tectonic reconstruction in carbonate rocks" (manuscript COMMSENV-24-1690-T). First of all, I sincerely apologize for the delay in sending my evaluation.

This manuscript investigates the potential of thermoluminescence (TL) dating on dolomite for application in thermochronometry. The authors provide a new series of data from the central Apennines as test for their novel approach. They propose a detailed analytical protocol to reveal the suitability of the TL dating on dolomite and its application for thermochronometry, with nice analytical results and high saturation doses that are highly valuable for slowly-exhuming regions compared to other OSL/TL techniques. The output TL ages reveal Late Pliocene exhumation history, that is placed in a broader context of the study region.

This is a very interesting manuscript, overall well-written and illustrated. It provided the first attempt to assess the potential of dolomite TL dating in thermochronometry, and as such this is a novel piece of work and important contribution with broad interest and potential of applications in the field of tectonics/geodynamics, especially in carbonate regions where other thermochronometric techniques are lacking. The detailed analytical results set the foundation of this novel technique, which would be further investigated to fully reveal the potential and possible limitations/complications. The application to the central Apennines is also interesting and provides quantification for the late Pliocene to Quaternary extensional tectonics associated to major earthquakes and hazards. Some of the output results and tectonic interpretations are not entirely clear and may be reworked or discussed for clarity. In addition, the significance of the TL ages as exhumation ages or rather related to direct fault activity (fluid circulation, re-crystallization or shear heating) should be better addressed in the discussion.

I have outlined below my questions and suggestions in a set of general and specific comments below.

General comments:

1 – Dolomite TL ages: the authors propose that TL ages on dolomite reflect the rock exhumation and associated cooling during fault activity. The dolomitization processes are exposed in the main text, but this is not entirely clear when this occurred compared to fault activity. In some places along the text it is mentioned that dolomite can be dissolved/recrystallized (joints? Fig. 2b), so I am wondering whether the obtained TL ages could alternatively represent other processes than rock exhumation in that context: (1) dolomite (re-)crystallization, or (2) potential re-heating during fault activity (shear heating or fluid circulation). Such processes have already been observed in other context and for other methods (U-Pb calcite, apatite U-Th/He, OSL/ESR...), so I think that this would be very interesting for readers to further discuss that in the main text and provide more observations/arguments to support the proposed interpretation of dolomite TL ages as constraints for rock exhumation.

Thank you very much for the comment! In the former manuscript, we only measured the alpha efficiency factors (S_a , the parameter for alpha dose rate calculation) of two clast dolomite samples. Their S_a values are very close to each other. So, we applied the mean S_a value of the clasts to calculate the alpha dose rate for dolomitized bedrocks as well. In order to have more precise constraints on the TL ages, we cooperated with another luminescence laboratory (Justus Liebig University of Giessen) to measure the S_a values of the bedrock samples (with repeated S_a measurements of the clasts). It turned out that the S_a values of bedrocks samples are much smaller than the clast samples. As a result, the TL ages of the two bedrock samples increase substantially, with the mean age increasing from ~3.4 Ma to ~4.6 Ma. Based on these updated TL ages, we have consequently revised our age interpretation substantially. The TL ages of bedrocks (4.8–4.4 Ma) indicate the syn-orogenic exhumation, whilst the TL ages of the clasts in the fault damage zone at 2.5 Ma indicate a signal resetting event as suggested by your comment. The REY (rare earth element + Y) signatures of the samples suggest a closed chemical system with no firm evidence of alteration/metasomatism by fluids. We suppose that the signal resetting event is most likely due to the frictional heating during coseismic faulting, which has also been reported for quartz and feldspar ESR and luminescence signals in previous studies. We therefore adapted the revised text to the updated dolomite TL age data set (see lines 193-220 and 234-251 of the revised manuscript).

2 – TL age difference clasts/bedrocks: there is 0.9 Ma age difference between the clasts and bedrock samples (line 103), but this difference is not really well explained in the main text. In Fig. 5 it seems that this is the result of westward propagation of the extension, but given the spatial proximity of the samples (distance/elevation) this seems questionable. Another alternative explanation could (possibly) be sample re-heating for the clasts by fault activity and shear-heating or fluid circulation (linked to my previous comment 1), could this be realistic?

Our former hypothesis was based on structural evidence, which indicates that the fault zone of the MMF is formed by multiple sub-parallel faults, suggesting a transition from delocalized to localized deformation.

However, based on the new TL age data, we changed our tectonic reconstruction (see lines 222-251 and the new Fig. 5 of the revised manuscript).

3 – Self-examination approach: I am not entirely sure that I follow and find useful the proposed self-examination approach proposed in the main text for cooling rate calculation (equ. 1, lines 119-126). I totally follow the author's modeling approach and T_c definition (trapping/detrapping balance) which is nicely represented by Fig. 3c. However, deriving an output cooling rate from equ. 1 appears somehow circular if compared to the output cooling rate from modeling (Fig. 4b), given that T_c is an output of the modeling approach (so the two cooling rate estimates are not independent). Maybe I missed something, please consider clarifying this or removing this point.

Thanks for pointing out this issue. Here we wish to show that the definition of T_c corresponding to a trapping/detrapping ratio of 0.5 is applicable. For example, if we define the T_c corresponding to a trapping/detrapping ratio of 0.1, the cooling rate calculated from age and T_c and surface temperature

will always be lower than the modeled cooling rate. We acknowledge that the wording here is not clear and misleading. In this revised manuscript, we have thoroughly revised this part. We calculated the mathematic T_c from the modeled cooling rate and the apparent age. Then we can find a trapping/detrapping ratio at the T_c from the trapping/detrapping vs Temperature relationship. It shows that this ratio is indeed ~ 0.5 for the samples. So we propose this alternative definition of T_c based on the trapping/detrapping ratio of 0.5. We have also simulated the change of T_c with different cooling rate and dose rates. The results are shown in the new Figure 4.

The new discussion is listed in lines 169-191 of the revised manuscript.

4 – Analytical results: I appreciate the extent of methodological details and data reported in the Methods section and Supplementary Materials. However, in the main text some of the important information may be missing for clarity or not really referenced, which is making sometimes difficult to follow the results and quality of data. For instance, would it be possible to show some of the TL spectra and MAAD dose-response curves in the main text for clarity (maybe one representative sample)?

Thanks for the suggestion. The Rev# 1 made a similar comment regarding the lack of essential information of the method. In the revised version, we have put more information in the main text. Moreover, Fig. 3 has been revised substantially, by adding the TL spectra, the TL glow curves and the MAAD dose-response curve (see lines 130-150 of the revised typescript).

Specific comments, by line number:

- Line 1. “Dolomite thermochronometry...”. I would suggest to add “TL” or “Thermoluminescence” for clarity in the title.

We have added ‘luminescence’ before ‘thermochronometry’ in the revised manuscript. We decided not to use the ‘thermoluminescence’, in order to avoid the repetition of ‘thermo’. And we don’t want to use abbreviation in the title.

- Line 19. “dolomite clasts in the fault zonr and dolomitized bedrocks”. See my general comment about possible (re-)crystallization of dolomite.

From the REY pattern of the clasts and matrix of the cataclasite, there is no firm evidence of alteration/metasomatism by fluids. Please refer to lines 213-217 of the revised typescript.

- Line 23. “exhumation/cooling history”. Are these TL ages reflecting really the cooling history, or rather a (re-)crystallization of re-heating event (fault shear heating)? See my general comment.

We agree that the TL ages of the clasts actually represent a reset event happened 2.5 Ma ago. We think that it is more likely that shear heating reset the signal, rather than the fluid-mediated (re-)crystallization based on the REY pattern. Please see the detailed discussion in lines 193-220 in the revised manuscript.

- Line 27. Maybe add section titles for clarity, such as “Introduction/Context” there.

Thanks for the suggestion. In the former manuscript, we did not add a header of ‘introduction’ at the beginning of the main text, following the layout of CEE. To make it convenient for review, we added a heading of ‘Introduction’ at the beginning.

- Line 29. “such as fission-tracks and (U-Th)/He methods...”. There have been some attempts to use U-Th/He dating on carbonates in the literature (e.g. Cros et al., 2014 GCA...), maybe provide some references there.

Thanks. We have added new lines 36-37:

(U-Th)/He dating has been explored for carbonates; however, its application is hindered by the low concentrations of U and Th, as well as the complex diffusion kinetics of helium within carbonate minerals (Crocs et al., 2014; Cherniak et al., 2015).

- Line 33. “to unravel the thermal/cooling histories of rocks 2-8”. I appreciate the amount of references cited there, but I would suggest to cite them progressively along the following lines as they do not provide the same information/success in trapped-charge application for thermochronometry.

Revised as suggested. Thanks.

- Line 42. “their luminescence and ESR signals”. Of what (quartz/feldspar)? Please correct.

Revised to ‘the luminescence signal (quartz and feldspar) and ESR signal (quartz)’ in line 62.

- Line 48. “structurally-controlled dolomitization”. Yes thus the TL ages on dolomite may also constrain (re-)crystallization and not necessarily cooling histories, especially in fault areas. See my general comment.

Yes we agree. Please refer to our reply above.

- Line 51. “its application in thermochronometry has never been explored”. The application of TL in calcite has been explored by Ronca and Zeller (1965), maybe this would be interesting for readers to compare the TL signal characteristics between calcite/dolomite.

Thanks. We have added these sentences (see lines 69-73 of the revised typescript):

‘Compared to dolomite, calcite is the more abundant carbonate mineral. The semi-stable 230 °C TL peak of calcite is sensitive to surface temperature and the thermal equilibrium state of this TL signal has been thus applied to reconstruct the past surface temperature on Earth (Ronca, 1964; Ronca and Zeller, 1965). This method is termed paleothermometry nowadays (Biswas et al., 2020). In contrast, dolomite has a strong TL peak above 300 °C, which has sufficiently high thermal stability at the surface temperature.’

- Line 53. “a strikingly high saturation dose”. Please provide a range/order of magnitude for readers.

We added ‘up to thousands of grays (Ramasamy et al., 2009; Akça-Özalp et al., 2021)’ in line 75.

- Line 54. “to be stronger than calcite”. To what? Mechanical deformation? Other? Please specify.

We have revised it to ‘Since dolomite rheology has been documented to be stronger than calcite during brittle deformation,’ in line 76-77.

- Line 61. “pre-orogenic dolomitized bedrocks”. I don’t know the study area, but given this information can the authors exclude potential re-crystallization during fault activity and associated fluid circulation? Same comment when reading line 82 “evidence of diffuse dolomitization”. Please clarify.

Thanks for the suggestion. The detailed discussion is in lines 193-220.

The REY signatures of the samples indicated a closed system. It seems that shear heating is more likely the cause of signal resetting, although we cannot exclude a contribution given by the structurally-controlled fluid-rock interaction in resetting the TL signals in the damage zone of the MMF.

- Line 69. “age of onset of the extensional tectonics”. Is there any study using calcite U-Pb dating on fault surfaces for this area? I would also strengthen there that beyond the onset age of fault activity, the aim of TL thermochronometry would be to provide exhumation rates indeed.

Thanks for the suggestion. We have included references to U-Pb dating in the central Apennines (see lines 101-102 of the revised typescript). We have stated that in the manuscript that the TL ages of the dolomitized bedrocks are cooling ages indicating the exhumation event.

- Line 89. “shear deformation”. Can fault shear heating may also influence the TL signal, as this has been reported/suggested for other luminescence signals (e.g. Bateman et al., 2012 QG, Lavé et al., 2023 Nature, Tsukamoto et al., 2024 Earth Planets & Space, Heydari et al., 2024 QG...)?

I think this is an important point to introduce in the text and to discuss in the lights of obtained results...

Indeed. In the revised manuscript, we highlighted that shear heating may play an important role in resetting the TL signals of the dolomite clasts. The ~2.5 Ma ages of the clasts indicate a seismic event 2.5 Ma ago which corresponds to the beginning of the post-orogenic extension (see lines 193-220 of the revised typescript).

- Line 92. “dolomite grains (present either as secondary mineral in the carbonate bedrock or clasts in cataclasites)”. This is unclear whether these investigated grains are inherited from pre-orogenic dolomitization of neofomed during fault activity. Please clarify and provide some arguments for this important point.

The grains are inherited from pre-orogenic dolomitization. We have revised the text in line 126-127: pre-orogenic dolomite grains (preserved either as carbonate bedrock or as fault clasts within the damage zone of the MMF)

- Line 96. There is no “Methods” section in the main text (this comes afterwards at the end), but maybe a quick sentence referring to the methodological section at the beginning of the results section could help the reader to follow.

Thanks for the suggestion. We have added the necessary information in the main text now.

- Line 101. “at different distances from the main fault traces”. How far? Please provide some numbers there (I have the impression that those samples are not that far from the faults, correct?).

They are located at ca. 20 and 100 meters away from the fault plane (see Fig. 1 and lines 132, 306 of the revised typescript).

- Line 113. “which the electron detrapping rate is half of the trapping rate”. Please refer to the methods section where these terms are defined.

We have moved the equation of charge trapping and detrapping into the main text now, in line 156. So the terms of electron trapping and detrapping have been introduced before this sentence.

- Lines 115-117. I don't really get the usefulness of calculating another cooling rate from the T_c (equation 1), which is only a rough estimate compared to the modeling approach you employed to derive cooling rates presented in the previous paragraph (lines 108-110). Please clarify or consider removing this (see my general comment on this).

This part has been thoroughly revised in the revised manuscript. Now we only discuss about the T_c . Please see the lines 170-192.

- Line 128. “50-70°C”. The range can even be larger, see Ault et al. (2019, Tectonics) for discussion and estimates.

Yes, indeed. In the revised manuscript, we simulated the T_c values based on different cooling rates (2-200 °C/Ma) and dose rates (0.1-2.0 Gy/ka). The range of cooling rates are the suitable application range of dolomite thermochronometry. The range of dose rates is wider than the true dose rates of dolomite samples in nature. Generally, the T_c increases with a higher cooling rate. The T_c does not change much with dose rate except when the cooling rate is lower than 5 °C/Ma.

These new contents are in lines 170-192 of the revised manuscript.

- Line 130. “a geothermal gradient of 25°C/km”. Are there some estimates for this area?

We have added the citation of Diaferia et al. (2019).

- Line 133. “showing a major erosion/exhumation event”. Please provide some rates there.

The rates are provided in the revised manuscript. Please see lines 246-248.

- Line 134. “the characteristic saturation dose (D_0) values range 2500–6000 Gy”. This is an important result, and could go before when giving the TL ages I think.

Thanks for the suggestion! In the revised manuscript, we prepared a separate section at the end of the main text, ‘The potential of dolomite TL thermochronometry’, to highlight the potential of the dolomite thermochronometry. We put the discussion of dating limit (high D_0 and low dose rate) into this section. Please see lines 253-298.

- Line 135. “taking a $2D_0$ as maximum”. I agree this is a standard calculation, at least in luminescence, but may add a reference there for readers (for instance Wintle 2008 Boreas?).

Added as suggested. Thanks.

- Lines 137. I would suggest to provide range estimates for maximum TL ages and associated cooling rates.

Thanks for the suggestion. We have calculated the maximum TL ages for each sample based on their individual D_0 and \dot{D} , and the maximum TL ages are in the range of 11-28 Ma. The cooling rate ranges for each sample were also calculated based on an n/N range of 5-85%. The minimum cooling rate is 1.7-4.1 °C/Ma, and the maximum cooling rate is 92-232 °C/Ma. These new contents are put under the section ‘The potential of the dolomite TL thermochronometry’, in lines 254-267.

- Line 143. “dolomite thermochronometry significantly broadens...”. How about potential fading, and/or variability in thermal kinetic parameters? I agree some of these are presented in the Methods section, but I would encourage the authors to discuss these further in the main text. This seems important when proposing a new approach.

We have added the related discussion in the section ‘The potential of the dolomite TL thermochronometry’. Please see lines 268-290.

- Line 147. “at the footwall of the MMF”. This fault system is complex, with multiple fault segments so can the exhumation/faulting history be more complex than presented in Figure 5?

Indeed. Based on the above reasoning and the new TL age data of the dolomitized bedrocks we have rethought the tectonic evolutionary model (see the new figure 5 and lines 222-251 of the revised typescript).

- Line 148. “were exhumed to near-surface conditions...”. Not correct, the samples were exhumed below the T_c but were still at 2-3km depth at that time, no?

Thank you for this comment, indeed is not correct, the TL age represents the time when the rock passes through the 60 °C temperature (2 km depth).

- Lines 155-160. I would disagree with these sentences and interpretation. No the Late Pliocene TL ages reported in this study do not constrain the onset time for exhumation/cooling, they quantify the time at closure depth/temperature but the exhumation could have been ongoing for longer time (samples were at greater depths). Please rephrase and correct this paragraph accordingly.

We agree. The TL age represents the time when the rock passes through the 60 °C temperature (2 km depth). We have revised accordingly. Please see lines 227-233.

- Line 162. “crustal extension progressed westward”. Is this evidenced by the TL age pattern, or combined with other data? Please clarify, from the TL ages the pattern is reduced (few hundreds of m) and not that clear, no?

With the updated ages of the dolomitized bedrocks, our interpretation has changed substantially. Please refer to lines 222-251 and the new Fig. 5.

- Line 169. “dolomite TL technique”. Replace by “dolomite TL dating”.

Revised as suggested.

- Line 169. “spatio-temporal variations in fault slip rates”. Unclear, the output results rather suggest no variations in fault activity. Maybe rephrase.

We agree. Please see lines 242-251 of the updated version of the manuscript.

- Line 173. “provides the only viable and effective approach...”. This sounds a bit strong to me and seems not necessary. The new dolomite TL ages presented there are interesting and highlight the potential of the method for thermochronometry, but I would suggest to rephrase and propose the potential (and stress the wide application) of this new method, to be confirmed by further analysis and case studies.

Revised as suggested.

- Line 174. “cooling/exhumation history”. Also for fault activity, no?

Yes. We have added a sentence in lines 296-298:

‘Finally, our study also indicates that the TL signal of dolomite in the fault zone can be completely reset during seismic events, and the corresponding TL ages can be used to constrain the paleo-seismic activity along major fault strands.’

- Line 178. “four dolomite clasts”. What is the size of samples?

We have added the information in the sample information section, lines 303-304: ‘The clasts are of sphere shape, with diameters of ca. 3–5 cm, and weights of ca. 40–200 g.’

- Line 180. “far away from the fault”. Please provide numbers there for distance to the fault.

We have added details in Line 305-307: ‘Two dolomitized bedrock samples were also collected for TL dating at different distances from the fault damage zone (~20 m for LUM4775 and ~100 m for LUM4774, respectively).’

- Line 181. Why only clasts and no bedrock were analyzed by XRD? Can this be possible that bedrock samples differ in mineralogy?

Thanks for pointing out this issue. We have added XRD analyses for the two bedrocks as well, together with five cataclasite matrix samples adjacent to the clasts. The results are displayed in revised figure S1. All of them are nearly pure dolomite.

- Line 189. “5mm thickness”. Is this enough to ensure no light exposure and TL signal removal for these outcrop samples? Usually for feldspar/quartz trapped-charge dating applied to thermochronometry, 1cm of the outer surfaces are removed (e.g. King et al., 2016 ChemGeol). Could this have an influence on the TL results, given the potential light penetration in clear dolomite/calcite crystals?

When we collected the clast samples, these clast samples were still inside the matrix of cataclasite. Only a small part of the surface was exposed. So the bleaching happened only at a small surface area of the surface. The clasts are often dark-grey in color and not transparent. After we collected the clasts, they were stored inside black plastic bags. In the laboratory, we used the 32 % HCl acid to remove about 40 % of the weight of the clast. This treatment should be enough to remove the partially exposed part. Since these clasts were of different sizes (diameters of ca. 3–5 cm, and weights of ca. 40–200 g), if there were still effects of partial bleaching, the degree of partial bleaching should be different between clasts of different sizes. However, all the clasts have quite similar TL ages clustered at ~2.5 Ma. This is also an evidence that there is not partial bleaching effect. For the dolomitized bedrocks,

they were of quite large sizes. We crushed them firstly into small fragments in dark room. Then, we used the small fragments from the middle part of the bedrock for HCl etching.

We have added more detailed information in the sample information section and sample preparation section of the revised manuscript, in lines 337-343:

‘To remove the effect of sunlight exposure, the clast samples were firstly immersed under the 32 % HCl acid for reaction. The volume of the HCl acid used of each clast was calculated to dissolve about half weight of the clast. Usually, after 1–2 h, the reaction became very weak. The clasts were taken out and washed by distilled water and weighed. The weight loss was generally 40 %. The two bedrock samples were large pieces of rocks. They were firstly crushed into small fragments under the subdued red light. The fragments from the middle part of the bedrock were chosen to be further reacted with the 32 % HCl acid to remove ~40 % of the weight.’

Please note that the description of ‘10% HCl for ~30 min’ in the former manuscript was a mistake. That is the preparation step for speleothem samples in the luminescence dating lab of LIAG.

- Line 191. The analysed grainsize is reported after sample preparation, but what is the initial grainsize for dolomite crystals? Have the authors made and looked at thin section to evaluate the dolomite grainsize? This can be much smaller, and implication for dose rate calculation?

Since the samples can be regarded as an infinite homogenous medium when calculating the alpha dose rate, the initial grain size has no influence on the dose rate calculation. We have added more related arguments in the lines 435-438:

‘The samples are almost pure dolomite from the XRD analyses, and thus an infinite homogeneous medium was assumed for dose rate calculation. The almost identical D_e values between the 63–100 μm and 100–200 μm fractions of LUM4525 (Table 1) further proves that the initial grain sizes of the dolomite crystals have no influence on the dose rate calculation.’

- Line 232. “peak shifting method”. Please provide reference(s) there. Same comment for line 238 (E and s values can be deduced...).

We added the reference of Hoogenstraaten (1958).

- Line 240. “Table S3”. I think that the estimated lifetimes should be discussed, first to state the relative stability of the TL360°C peak, and also I would encourage the authors to discuss the potential usefulness of the TL285°C (lower thermal stability, so complementary to the other peak for tighter constraints on the late-stage cooling history), although practically at present this peak cannot be used (but maybe future work will allow to use it?). Please consider adding few lines about this.

Thanks for the suggestion. We have added the some discussion on the 285 °C TL peak, in the new section ‘The potential of the dolomite TL thermochronometry’, in lines 268-290.

The low thermal stability of the 285 °C TL peak means low closure temperature, which has potential to provide constraints on the late-stage cooling history. However, the low thermal stability show means that the signal saturation level at the equilibrium state, $(n/N)_{ss}$, will also be lower than 1. If the $(n/N)_{ss}$ is too small (e.g. 0.15 for bedrock LUM4774), it will restrict the application of thermochronometry. In this case, this signal might be more suitable for paleothermometry.

We have added a supplementary in the revised manuscript for clarification.

Fig S9:

- Line 252. I am not completely convinced by the fading results presented in figure S7. First there seems to be an apparent peak shift measured in the 360°C region, is it just a visual impression or real? And in addition, would it be possible to present the fading results centered on the region of interest, i.e. the 340-380°C region, to better evaluate the results? In any case, I would encourage the authors to add few words in the main text about this, as well as for the thermal kinetic parameters.

To have a quantitative estimation of fading rates (g -values), we have added experiments with the SAR TL protocol for two samples and their g -values are close to zero. Also, for the measurements this time, the TL curves measured promptly after irradiation and after 9 days of irradiation did not show peak shift. We suspect that the TL peaks shift in the original Fig S7 (new Fig S8a) might be related to the poor reproducibility of the luminescence reader at that time.

We added some discussion about fading and thermal kinetics in lines 268-290.

- Line 255. Where are the U-Th-K contents reported? Please provide the data in text or as table.

A new table S5 was added with the details of U, Th, K concentrations, as well as the S_a values.

- Line 290. “the samples were far below the surface”. How far? In the sample collection section, these are reported to be sampled on the fault plane, not at depth. Please clarify, given that cosmic rays can penetrate as deep as meters this may not be negligible to consider this effect on sample dose rates.

Yes, the samples are now at the surface. However, for most of the time within the cooling age, the samples are far below the surface. We have calculated the cosmic ray dose rate with depth at the L’Aquila site. When the depth is more than 35 m, the cosmic ray is less than 0.01 Gy/ka. We have added the arguments in the ‘Dose rate calculation’ section in lines 478-482 and added a supplementary figure 13:

‘Cosmic ray dose rate decays exponentially with depth (Prescott and Hutton, 1994), and it is lower than 0.01 Gy/ka once the depth is larger than 35 m (Fig. S13). With an exhumation rate of 0.45 mm/year, a depth of 35 m corresponds to an age of 78 ka. Considering the high ages of the samples, the contribution of cosmic ray to the total dose rate is negligible. Thus the cosmic ray is not taken into consideration for dose rate calculation.’

Fig S13:

- Line 293. “assuming first-order kinetics”. Is this a valid assumption? From Fig. S6 I would say yes, but this should be discussed and possibly are there any references supporting this point?

In the revised manuscript, we have modeled the cooling rate for one bedrock sample (LUM4774) with a multiple first-order kinetic model assuming a Gaussian distribution of trap depths.

We find that the cooling rate decreases a little bit with a higher standard deviation of the activation energy (σE). Considering the errors, they are still consistent with a single first-order trap model ($\sigma E = 0$). These results are added into the ‘Cooling rate modeling section’ in lines 496-505, and the Fig. S14.

Fig. S14:

We have not tried the cooling rate modeling with non-first order kinetics. With non-first order kinetics, the equation (2) in the manuscript for the charge accumulation will change. Nevertheless, an advantage of the peak shifting method is that the activation energy (E) can still be accurately estimated, irrespective of the order of kinetics (Singh et al, 1990).

- Line 295. How is n/N defined from the measurements? In table 1 this is defined as the “natural TL signal intensity”, but this is not clear. It should be linked somehow to the parameters of the dose response curve fitting (Fig. S3) but no indication are given, please clarify.

Thanks for the suggestion. We have added sufficient details regarding how the signal saturation level is determined in the revised section ‘Dolomite TL ages and cooling rates’ in lines 130-144, and made links to the dose response curve in the new Fig 3c.

- Line 295. This is not clear that equation 1 is in fact 2 equations next to each other, maybe consider showing them on two lines for clarity.

We think it might be better to put them into one equation. In this way, it will be more clear to the readers that the trapping probability equals to \dot{D}/D_0 and the detrapping probability equals to $1/\text{lifetime}$. Also we have moved this equation to the main texts to make it easier for the readers to follow the method. See line 156.

- Table 1. Please provide a clearer definition of n/N , and refer to text and equations for T_c (“with our definition” is not clear).

Revised as suggested. Thanks!

- Figure 2. In panel a, what are the letters a,b,c on the block diagram? In the stereoplots, the colors are not clear on my version, maybe try to update them for clarity.

On panel d, is it possible to have the sampled surface indicated on the picture?

Thank you for noticing it. The letters correspond to the panels at the bottom, they should be b, c and d and not a, b and c. The stereoplots are not clear due to the export in jpg, the pdf version are clear, we tried to improve the clarity in the jpg as well. In panel d we added the samples.

- Figure 3. On panel c, what are the different temperature-lines shown? Every 10%? Please describe this in caption, and maybe indicate the 10-50-90% lines in bold. In caption, the panel a and b differ (cooling history modeling vs. cooling rate modeling), please correct for consistency.

Yes, on the panel c of former figure 3, it is an increment of 10 %. We have indicated them with numbers in the revised figure 3f. We have revised Fig 3 substantially, by adding more information about the TL dating method.

- Figure 4b. See my general comment, I am not fully convinced by the self examination procedure and results reported in Fig. 4b.

Figure 4 is replaced by a figure of T_c in the revised manuscript.

- Figure 5. Some comments/suggestions for this figure: The isotherm showed on the figure is not correct, it should first follow the topography and is perturbed by the fault advection. Please correct.

In panel a, no exhumation is considered but the data cannot constrain this (since rocks are above the T_c), so there might be exhumation/fault activity already before no?

In panel b, what is the timing and importance of the thrust cataclasite? This is not discussed in the main text...

Finally, in panel c is illustrated the fault propagation to explain the ~0.9Ma difference in TL ages. Is it the case (I could not figure out this explicitly in the text, or i missed it) and is it realistic given the proximity in samples (both in distance and elevation)?

To reconcile the new TL data within the geological setting of the study area, we propose a new evolutionary model described in lines 222-251 of the revised manuscript, and illustrated in Figure 5. Exhumation of the AIB shoulders occurred in the Early Pliocene and during the syn- and post-orogenic phase.

I hope these comments and suggestions may be useful for revising this interesting manuscript.

Sincerely,
Pierre Valla
(Grenoble, 26 July 2024)

Reviewer #3 (Remarks to the Author):

Comments to the manuscript:

Title: Dolomite thermochronometry applied in central Apennines: a novel tool for tectonic reconstruction in carbonate rocks.

Authors:Zhang et al.

The authors present the results of a new thermochronological tool to assess the exhumation/cooling history of carbonate rocks based on the dolomite thermoluminescence (TL) dating.

As claimed by the authors, the results of these studies could have a very important impact on the reconstructions of the exhumation/cooling history in carbonate rock regions which until now have suffered the lack of appropriate thermochronological methods due to the absence of target minerals (e.g., zircon, apatite, quartz, feldspar). Furthermore, since the TL signal of dolomite has a strikingly high saturation dose, the method could bring a great advantage in thermal history reconstruction in regions with low cooling rates (i.e., as low as 5 °C/Ma), also providing a useful tectonic/thermal marker to investigate long-term crustal deformation in the brittle seismogenic crust.

Therefore, the manuscript represents a novel and valid contribution to the Earth Sciences community and is eligible for Communications Earth & Environment journal.

The various sections presented in the “Methods” chapter are written in a very thorough and detailed way. However, I believe that a very concise sentence at the end of each section could help even readers who are not specialized in the techniques used to understand the fundamental aspects of the method, the various critical issues encountered and how they were overcome.

Regarding the case of application of the novel thermochronometry method in the Central Apennines, three main points of criticism arise, which I believe should be considered to make the work suitable for publication in “Communications Earth & Environment”.

1) My main criticism is aimed at the fact that the exhumation values evaluated for the Monte Marine area from the Late Pliocene to Present day are always discussed with regard to the MMF activity alone, so much so that throughout the manuscript the presence and contribution to the exhumation of rocks due to the "regional uplift" of the Apennines is never discussed.

Considering for example the very recent paper by Racano et al. (2024), where there are also common co-authors with the present manuscript, it is argued that: ... “Thermochronological and geochemical studies provide evidence for the recent rapid uplift in the Central Apennines. Apatite fission track and (U-Th)/He cooling ages in the Central Apennines reveal that the most recent exhumation phase began at approximately 2.5 Ma, coinciding with the onset of normal faulting and the opening of intra-montane basins throughout the Central Apennines (Cosentino et al., 2017; Fellin et al., 2021)”.

I believe, therefore, that it is necessary to indicate in the Introduction section also the presence of the contemporary crustal process of the uplift of the Central Apennines, and perhaps discuss in the next part of the manuscript the possibility of discriminating and evaluating the extent of exhumation associated with the local activity of the MMF compared to the regional uplift of the Central Apennines.

Thanks a lot for your comments! We welcome these observations and we agree that Introduction and discussion should be modified accordingly (see lines 92-105, and 222-251 of the revised typescript). As stated above, the new dolomite TL ages have determined to propose a new evolutionary scenario, where the TL ages from the fault clasts must be interpreted as the age of coseismic faulting rather than cooling/exhumation ages.

Still regarding the aforementioned issue, I also ask whether the hypothesis of analyzing a dolomite sample (if present) also in the hanging wall block of the MMF has been evaluated, considering that the bedrock is in outcrop both towards the most northwestern and southeastern portions of the fault (e.g., CARG Sheet 348 “Antrodoco”, ISPRA, at https://www.isprambiente.gov.it/Media/carg/348_ANTRODOCO/Foglio.html).

Although, we believe that this would be really interesting, unfortunately, this is not possible because the hanging wall of the MMF is composed of siliciclastic rocks (Marne con Cerrognana) and by the foredeep deposits. The only dolomite rocks are present at the footwall of the MMF.

2) The authors argue that the exhumation rates obtained through the dolomite TL dating (up to 0.8–0.9 mm/yr, see line 163-167) are compatible with the paleoseismic estimates of fault slip rates in the central Apennines, which are in the order of 0.4–1.2 mm/yr (preferred 0.6–0.8 mm/yr). However, I suggest that it might be more correct to compare the exhumation rate obtained by dolomite thermochronometry with the "downthrow-rate" associated with the MMF, especially considering that the main fault plane often exhibits a "quite low-angle" geometry (i.e., 40–45°, in the Barete area) as it seems to be observed from the stereoplot of fig. 2a.

Following the reviewer’s advice, we have now used the stratigraphic separation across the MMF to obtain the maximum long-term slip rate along the MMF, adopting an average fault dip of 65°. Indeed, we performed a statistical analysis on the normal fault planes (n = 320) along the MMF, which indicates a dip angle of $67^\circ \pm 9^\circ$, which aligns with the expected dip of a normal fault. Transferring the exhumation depth to fault slip, we obtain a maximum, long-term slip rate along the MMF of ca. 0.75 mm/yr. Please see lines 234-251 in the revised manuscript.

3) In the schematic 2D evolutionary model of Fig. 5 it is shown that from about 3.4 to about 2.5 Ma (so at least for about 0.9 Ma) only the easternmost fault plane of the MMF is active - that is, the one between the TL SAMPLE (bedrock) and the TL SAMPLE (Cataclasite), then faulting progressed westward post c. 2.5 Ma. In this hypothesis, considering a "slip-rate" of about 0.6-0.8 mm/yr (see line 167) this segment should have accumulated a slip of 500-700 m. Do the authors have geological evidence in the field of this easternmost plane with such values of cumulative slip?

Thank you for this comment. As explained above, the new TL age data set have forced a rethinking of the evolutionary scenario, where the ca. 2.5 Ma dolomite TL age of the fault clasts are framed in a scenario of coseismic faulting at the onset of the D2 post-orogenic extension in the region. Please see the arguments in lines 193-220, and the new figure 5.

You should feel free to use my comments as they feel appropriate. I hope they are useful. Thank you for considering me as a reviewer for this manuscript.

Very Truly Yours

References:

Gruppo di Lavoro, M. S., 2008. Indirizzi e criteri per la microzonazione sismica. In Conferenza delle Regioni e delle Province autonome-Dipartimento della protezione civile, Roma (Vol. 3).

Hoogenstraaten, W., 1958. Electron traps in zinc sulphide phosphors. Philips Research Report 13, 515–693.

Medlin, W.L., 1961, Thermoluminescence in Dolomite: The Journal of Chemical Physics, v. 34, p. 672–677, doi:10.1063/1.1701008.

Prescott, J.R., Hutton, J.T., 1994. Cosmic-Ray Contributions to Dose-Rates for Luminescence and ESR Dating - Large Depths and Long-Term Time Variations. Radiat Meas 23, 497–500. [https://doi.org/10.1016/1350-4487\(94\)90086-8](https://doi.org/10.1016/1350-4487(94)90086-8)

Singh, T.S.C., Mazumdar, P.S., Gartia, R.K., 1990. A critical appraisal of methods of various heating rates for the determination of the activation energy of a thermoluminescence peak. J. Phys. D: Appl. Phys. 23, 562–566. <https://doi.org/10.1088/0022-3727/23/5/014>

Response to reviewers' comments on **COMMSENV-24-1690A**, entitled '*Dolomite luminescence thermochronometry: a novel tool for tectonic reconstruction in carbonate rocks with application to central Apennines*', by Junjie Zhang, Giorgio Arriga, Federico Rossetti, Valentina Argante, Dennis Kraemer, Mariana Sontag-González, Domenico Cosentino, Paola Cipollari, Sumiko Tsukamoto

Dear editors and reviewers,

Thanks a lot for your time and efforts for reviewing our manuscript! We have revised our manuscript following your comments. Below, we provide a detailed point-by-point response to the comments. To make it convenient for reading, we make our reply in blue.

REVIEWERS' COMMENTS:

Reviewer #2 (Remarks to the Author):

Dear Authors, dear Editors,

Please find below my evaluation concerning the revised manuscript by Zhang and co-authors entitled "Dolomite luminescence thermochronometry applied in central Apennines: a novel tool for tectonic reconstruction in carbonate rocks".

I thank the authors for the thorough revisions on the manuscript, which have answered most of my initial comments. I think that the revised manuscript is clearer, especially regarding the proposed methodology and TL approach for thermochronometry and fault activity, which is the central and novel piece of the present work. I still have few comments regarding the data/modeling presentation and discussion of the output results for tectonic reconstruction. My main concern is about the use/referencing of "Methods" section and of the Supplementary Material information in the main text, which is lacking sometimes and prevent the readers to easily follow the methodology or the results. Concerning the tectonic reconstruction, I appreciate the discussion of TL outputs for clasts and the possible interpretation as reflecting faulting events. However, I still think the proposed interpretation (one single reset event) is a strong hypothesis and would require more discussion.

I have outlined below my questions and suggestions in a set of specific comments.

Many thanks for your constructive comments for our manuscript. We have revised the method section and supplementary materials to increase the readability. We have added more discussion regarding the age interpretation of the fault clasts. We think that 'complete resetting' of the TL signals of fault clasts is needed to explain their consistent TL ages, considering the different sizes and locations of the clasts. The 'complete resetting' can be accomplished by one major seismic event, or by several seismic events within a seismic phase. Regarding your suggestion that the TL ages of clasts could also be an apparent result of 'several small resetting events that have accumulated over the exhumation history of the clasts', we agree that theoretically there is this possibility. However, the consistent TL ages of the clasts cannot be easily explained by multiple partial resetting events as that will require similar degrees of partial resetting for the clasts of different sizes and locations, which is highly unlikely. We have added related discussion in the revised manuscript. The detailed replies to each individual comment are listed below.

Specific comments, by line number:

- Line 17. “the cooling and exhumation history”. Maybe replace history by histories.

Corrected as suggested.

- Line 20. “novel thermochronometric technique”.

Corrected as suggested.

- Line 21. “The study area is located...”. Maybe rephrase this sentence beginning to stage that you apply the novel approach to a specific area...

We have revised it in the new line 19:

‘Thermoluminescence dating is applied along the northeastern shoulder of the Late Pliocene-Quaternary L’Aquila Intermontane Basin, at the footwall of the extensional Monte Marine Fault.’

- Line 48. “The growth of the luminescence signal...”. Maybe a reference is missing there.

A reference of Wintle (2008) is added.

- Line 56. “luminescence dating”. “thermochronometry” would better apply there. And same comment for line 58.

Revised as suggested.

- Line 62. “In regions with low exhumation rates”. What range is considered there? And above in the text are reported cooling rates, not exhumation rates. Please correct.

This sentence has been revised in new lines 62-64

‘In regions with cooling rates lower than $200\text{ }^{\circ}\text{C Ma}^{-1}$ (with regard to luminescence) or lower than $20\text{ }^{\circ}\text{C Ma}^{-1}$ (with regard to ESR), the luminescence signals of quartz and feldspar and the ESR signal of quartz in rocks exhumed to the surface are already in saturation’

- Line 65. “to rock bodies”. Maybe “lithologies” or “bedrocks” would be better there, no?

Revised to ‘bedrocks’ as suggested.

- Line 73. “This method is termed paleothermometry nowadays”. Is this sentence useful for the message?

This sentence is removed.

- Line 74. “a strong TL peak above $300\text{ }^{\circ}\text{C}$,...”. Please provide a reference there.

Reference added.

- Line 74. “at the surface temperature”. Would this be valid in hot settings, i.e. deserts?

According to our modelling, the thermal stability is sufficiently high at surface temperatures $< 30\text{ }^{\circ}\text{C}$. But when the surface temperature is higher than $30\text{ }^{\circ}\text{C}$, the signal at the field saturation state is lower than 1 (Fig 3D). So, we have revised this sentence in new lines 73-74:

‘In contrast, dolomite has a strong TL peak above $300\text{ }^{\circ}\text{C}^{22-24}$, which has much higher thermal stability at the surface temperature.’

- Line 94. “in the transition”. Maybe “during the transition” would read better there.

Revised as suggested.

- Line 95. “D1 orogenic”. The deformation phases D1 and D2 are only presented after in the next section (line 119 and Figure 2), so I would suggest to rephrase and be more explicit there.

Revised to ‘The orogenic crustal shortening regime (stage D₁)...’ and ‘The onset of the post-orogenic extension (stage D₂)...’, in new lines 95 and 98.

- Line 133. “Four dolomite clasts”. Maybe refer to the Supplementary Material and/or Methods section for sample location (Table S1, etc).

We have added ‘(Fig. 1c; Table S1; Methods section)’ after this sentence.

- Line 142. “In case of dating,...”. I am not sure the term “dating” is clearly applicable there (same remark for line 143 and later in the text), since the purpose is not really dating but thermochronometry...maybe check and rephrase.

We revised this sentence in new line 141-142:

‘In thermochronometric application, TL curves with laboratory doses were measured after a preheat to 260 °C to remove low-temperature peaks, and the 350 °C TL peak is of our interest.’

- Line 144. “A multiple-aliquot additive-dose dating protocol”. A reference is missing there.

The reference of Wintle (1978) has been added here.

Wintle, A.G., 1978, A thermoluminescence dating study of some Quaternary calcite: potential and problems: Canadian Journal of Earth Sciences, v. 15, p. 1977–1986, doi:10.1139/e78-208.

- Line 149. “TL ages”. I would recommend to add “apparent” before age since this is a closure age (as discussed after).

Revised as suggested.

- Line 154. “For trapped charge dating”. I would suggest to replace dating by thermochronometry.

Revised to ‘trapped charge thermochronometry’ as suggested.

- Line 183. “E varies in the range of 1.71–1.88 eV, logs in the range of 13.0–14.6, and D0 in the range of 2450–5920 Gy.”. Please refer to Supp. Material and figures for these results.

These information are listed in Table 1. We have added ‘(Table 1)’ at the end of this sentence, in new line 183.

- Line 185. “a medium E of 1.80 eV”. What is a “medium” value? Average? Median? Please clarify.

It is a ‘median’. We have corrected the word. Thanks.

- Line 189. As for my initial review, I still don’t see the point of calculating two T_c values, since they are similar (T_c is extracted from the cooling rate, and corresponding roughly to T_{c_half} as written on lines 177). So there is no real meaning to compare the two (Fig. 4b), or I have missed something...please clarify or remove this sentence and comparison.

Revised as suggested. We have deleted the discussion about Tc_half and the comparison figure of the two Tc values (original Fig. 4b).

- Line 201. “This evidence suggests a closed chemical system”. Yes I agree the patterns are similar between bedrocks and clasts in Fig. S6, but the clasts are clearly enriched in all REY so would it be plausible to have an homogenous enrichment? Please clarify.

Thanks for pointing out this issue. We think that the variations of the overall REY concentrations between the samples are due to regional variations and it is a coincidence that the two bedrock samples we dated are from a region which has low REY concentrations.

In a related ongoing study, we have analyzed REY data for 7 bedrock samples from the Monte Marine Fault, and their REY concentrations vary substantially, with some samples higher than the clasts in cataclasites. And, the samples still have the consistent fractionation patterns (Figure below).

Figure in response: REY data of samples from Maonte Marine Fault (unpublished data).

A previous study on the Italian limestones also reported REY concentrations variation up to three orders of magnitude, with no signs of hydrothermal alteration (Rosatelli et al., 2023).

Rosatelli, G., Castorina, F., Consalvo, A., Brozzetti, F., Ciavardelli, D., Perna, M.G., Bell, K., Bello, S., and Stoppa, F., 2023, Elemental abundances and isotopic composition of Italian limestones: Glimpses into the evolution of the Tethys: *Journal of Asian Earth Sciences*: X, v. 9, p. 100136, doi:10.1016/j.jaesx.2023.100136.

We have added the related discussion in new lines 197-202:

‘Variations in the overall REY concentrations are probably related to regional variations within the lithological unit, as the sample sites are hundreds of meters apart. Indeed, previous studies have reported REY concentrations variation up to three orders of magnitude for Italian

limestones⁷¹. The similarity of the REY fractionation patterns of the studied samples suggests a closed chemical system without significant alteration/metasomatism in the carbonate bedrock during faulting.'

- Lines 207-210. "TL ages of the fault clasts are nearly coincident, regardless of their sizes and the along-strike locations within the fault damage zone. Thus, the ~2.5 Ma TL ages of the dolomite fault clasts likely indicate a complete resetting event induced by shear heating during development of the MMF".

I would somehow disagree with this strong statement about the significance of the 2.5Ma TL age for clasts. First, if the block exhumation is similar to the bedrock samples, we need to consider that at ~2.5 Ma the clasts were also still at depth. With the bedrock cooling rate of ~11°C/Ma, this relates to ~37.5°C so the thermal detrapping rate may not be zero. From Figure 2f this can be small, but this first needs to be evaluated/discussed by authors.

Thanks for pointing out this issue. We have added these discussion in new lines 221-227:

'With a cooling rate of 11.3 °C Ma⁻¹, a TL age of 2.5 Ma corresponds to a geothermal temperature of 38 °C, under which the thermal detrapping/trapping rate ratio of the TL signal is smaller than 0.05 (Fig. 4c). Additionally, the TL signals' thermal lifetimes at 35 °C of the clasts are calculated to be 28–99 Ma, which are already more than 10 times longer than the 2.5 Ma age. The thermal lifetimes at lower temperatures would even longer. Thus, the TL signal can be regarded as sufficiently stable at temperatures under 35 °C, and thermal loss of the TL signal is negligible. The apparent TL age of ~2.5 Ma should represent the true age when the resetting happened.'

Second, I agree that the overall very similar age around 2.5 Ma for all clasts is a nice results and should be better emphasized as reflecting a similar trapped-charge history for these samples, but this could also be several small resetting events that have accumulated over the exhumation history of the clasts...this needs to be better discussed I think.

We thank the reviewer for this advice. We have therefore modified the pertinent section in the discussion taking into account and discussing the multiple events hypothesis, in new lines 206-215:

'Since polyphase and active seismogenic faulting is documented along the MMF since Pleistocene^{36,56,78,79}, the apparent TL ages of ~2.5 Ma of the fault clasts might be a combined result of several partial resetting events that accumulated over the exhumation history of the clasts. However, under this scenario, the similar apparent TL ages of the fault clasts would require similar degrees of partial resetting between the clasts during these events. Considering the different sizes and different along-strike locations of the clasts which should result in different susceptibilities to frictional heating, the partial resetting with similar degrees of signal loss between different clasts is not a feasible scenario. Instead, the similar TL ages of the clasts can be reasonably explained by a complete resetting scenario, which can be achieved by a single seismic event or multiple seismic events within a seismic phase, as long as the frictional heating is sufficiently intense.'

Also in new lines 227-230:

‘Given these considerations, we hypothesize that the TL signals of the fault clasts were reset completely during a major seismic event (or a major seismic phase with multiple seismic events) at ~2.5 Ma ago, likely associated with the onset of D2 post-orogenic extensional faulting in the central Apennines.’

- Line 232. “an average exhumation rate of 0.45 mm/yr”. Please propagate the uncertainties on cooling rates for the exhumation rates. This will allow the readers to better compare with literature estimates.

Thanks for the suggestion. We have modified this part in new lines 239-243:

‘From the cooling rates of the two bedrock samples (11.3 ± 2.2 and 11.3 ± 1.7 °C Ma⁻¹), the exhumation rates are calculated to be 0.45 ± 0.09 and 0.45 ± 0.07 mm yr⁻¹, respectively. The mean exhumation rate is 0.45 ± 0.06 mm yr⁻¹. It indicates that the carbonate bedrock travelled through a paleo-geothermal temperature of 60 °C (2 km depth) at 4.60 ± 0.35 Ma and continued to be exhumed to the surface with an average exhumation rate of 0.45 ± 0.06 mm yr⁻¹.’

- Line 237. “average: ~2.5 Ma”). What is the uncertainty that can be derived from all the clasts? This could then be propagated to the slip rate estimate, no?

Thanks for the suggestion. The modified context is in lines 254-260:

‘After the signal resetting at 2.53 ± 0.13 Ma, ...a maximum long-term (tectonic- and erosion-related) exhumation rate of 0.71 ± 0.04 mm yr⁻¹ can be thus derived for the footwall carbonate bedrocks, converting to an average maximum fault slip rate of 0.78 ± 0.04 mm yr⁻¹ (Fig. 5c).’

- Line 238. “indicate a fault-related signal resetting event”. See my comment above for lines 207-210. This apparent age can also be the sum of small multiple events, so the link between onset of D2 and this apparent age is a strong hypothesis. I would recommend the authors to tune down this and/or explain why they think this hypothesis could be valid (independent observations or data?).

We thank the reviewer for this advice. As stated above, we modified as requested in the discussion section (please refer to lines 206-230 of the revised typescript).

We think that the resetting could be one complete resetting event, or multiple resetting events within a short seismic phase that achieved complete resetting as well. However, the consistent TL ages of the clasts are unlikely to be explained by multiple partial resetting events as that will require similar degrees of resetting for clasts of different sizes and locations, which would be an improbable coincidence.

- Line 814. Figure 3. There is a lot of information on this figure, panels are small and we cannot read text in panel c. I would recommend the authors to split this into two figures, one for modeling (d-e-f) and one for TL analysis (a-b-c).

Revised as suggested.

- Line 834. Figure 4. As stated above, I don’t think that figure 4b is insightful, compared to the figure 4a is important and valuable.

Original fig 4b is deleted as suggested.

- Supplementary material: This document compiles a lot of technical information that are important for the study and TL thermochronometric approach, but there is no real structure and only figures and tables are provided, which may not help the readers. Maybe re-structure the document with different sections, starting with sample information, TL analysis, modeling etc. At present this is difficult to follow it for (non-expert) readers...

We have revised the supplementary material as suggested. Now, the order of supplementary figures has been changed, and the figures are divided into different groups, with supplementary note in each group to give introduction for these figures.

I hope these comments and suggestions may be useful for finalizing the revisions for this manuscript, and that they could make the authors message clearer.

Sincerely,

Pierre Valla

(Grenoble, 5 February 2025)

We really appreciated the reviewer's constructive comments and suggestions which have helped to increase the readability and clarity of our manuscript greatly.

Reviewer #3 (Remarks to the Author):

Dear Editor and dear Authors,

I read with pleasure the revised manuscript titled "Dolomite luminescence thermochronometry: a novel tool for tectonic reconstruction in carbonate rocks with application to central Apennines" by Dr Zhang and colleagues.

I really appreciated the Authors' effort in considering most of the comments made by the reviewers. I believe that this has brought a concrete benefit not only in the presentation of the innovative geochronological method, but also in its application example to the tectonic context of the Apennines.

With respect to this last point, the new evolutionary model now proposed that, thanks also to the new TL dolomite ages, indicates a coseismic signal resetting event at c. 2.5 Ma, most likely due to the frictional heating at the onset of the D2 post-orogenic extension (rather than cooling/exhumation ages), provides a very stimulating hypothesis that also addressed my first question about the problem of discerning the exhumation associated with the local activity of the MMF compared to the regional uplift of the Central Apennines.

Congratulations, also, about the new figure 5 that schematizes in a very clear and effective way the results of the study.

Finally, concerning the compatibility of the MMF slip rate values obtained in the present work with the short-term ones (lines 246-248), I suggest the authors also consider the recent paleoseismic study by Iezzi et al. (2023) who estimated an average slip rate value of about 1.0 mm/yr for the last 35 kyr of MMF fault activity.

Iezzi, F., et al. (2023) - Slip localization on multiple fault splays accommodating distributed deformation across normal fault complexities. TECTONOPHYSICS. - ISSN 0040-1951. - 868:, p. 230075. [10.1016/j.tecto.2023.230075]

Therefore, I hereby inform you that I am fully satisfied with the Authors' response and the revised version of the manuscript.

I hope that my contribution has been useful to the Authors and thank you again for considering me as a reviewer for this manuscript.

Best Regards

Thanks a lot for your comments and suggestions which have greatly improved the quality of our manuscript. In the revised version, we have added the new reference of Iezzi et al. (2023) in the discussion part regarding the slip rate.